# Understanding Representation of Deep Equilibrium Models from Neural Collapse Perspective

**Haixiang Sun**
ShanghaiTech University
sunhx@shanghaitech.edu.cn

**Ye Shi** *
ShanghaiTech University
shiye@shanghaitech.edu.cn

## Abstract

Deep Equilibrium Model (DEQ), which serves as a typical implicit neural network, emphasizes their memory efficiency and competitive performance compared to explicit neural networks. However, there has been relatively limited theoretical analysis on the representation of DEQ. In this paper, we utilize the Neural Collapse ($\mathcal{NC}$) as a tool to systematically analyze the representation of DEQ under both balanced and imbalanced conditions. $\mathcal{NC}$ is an interesting phenomenon in the neural network training process that characterizes the geometry of class features and classifier weights. While extensively studied in traditional explicit neural networks, the $\mathcal{NC}$ phenomenon has not received substantial attention in the context of implicit neural networks. We theoretically show that $\mathcal{NC}$ exists in DEQ under balanced conditions. Moreover, in imbalanced settings, despite the presence of minority collapse, DEQ demonstrated advantages over explicit neural networks. These advantages include the convergence of extracted features to the vertices of a simplex equiangular tight frame and self-duality properties under mild conditions, highlighting DEQ's superiority in handling imbalanced datasets. Finally, we validate our theoretical analyses through experiments in both balanced and imbalanced scenarios.

## 1 Introduction

Recently, there has been significant research on *implicitly-defined layers* in neural networks [1, 2, 3, 6, 9, 20, 21, 49], where the output is implicitly mapped from the input under certain conditions. These layers embed interpretability and introduce inductive bias [26] into black-box neural networks, demonstrating superior performance compared to existing explicit layers.

Among these implicit networks, the Deep Equilibrium Model (DEQ) is a memory-efficient architecture that represents all hidden layers as the equilibrium point of a nonlinear fixed-point equation. Due to the absence of a closed-form solution in its forward process, DEQ can be viewed as having an infinite number of layers during iteration as long as the threshold is set low enough, enhancing its ability to fit input data. Consequently, its representational capacity is relatively stronger compared to a single-layer network structure. This phenomenon explains why DEQ has achieved state-of-the-art results in classification tasks compared to existing architectures like ResNet. For instance, it has been successfully applied to language tasks and image classification tasks, reaching state-of-the-art performance. Additionally, DEQ can be applied in various domains and integrated with numerous other models, including inverse problems [19], Neural ODEs [11], diffusion models [24, 43], Gaussian processes [17], and more.

However, recent research reveals a phenomenon called Neural Collapse ($\mathcal{NC}$) concerning the learned deep representations across datasets in image classification tasks [42]. Under the $\mathcal{NC}$ regime, the last-

---

*Corresponding author.

38th Conference on Neural Information Processing Systems (NeurIPS 2024).

layer feature of each sample in neural networks collapses to their within-class mean, and the classifier vector converges to a simplex Equiangular Tight Frame (ETF). Theoretical analyses [10, 40, 51] indicate that under the Unconstrained Features Mode (UFM) condition, specific features $\boldsymbol{H}^0$ can be isolated from the entire network, known as the layer-peeled model [14]. In this scenario, Neural Collapse ($\mathcal{NC}$) is observed under certain conditions, suggesting that $\mathcal{NC}$ is agnostic to the backbone of feature extraction. Moreover, since $\mathcal{NC}$ measures the degree of proximity between features of the same category, an imbalanced dataset can exert a more negative influence on the performance of $\mathcal{NC}$. For instance, classes with fewer samples may not separate well and could converge in the same direction, leading to what is known as *Minority Collapse* [14]. Thus, the $\mathcal{NC}$ metric serves as a valuable indicator of a model's behavior in the context of imbalanced datasets.

The reasons behind the superior performance of DEQ still lack theoretical proof and comprehensive quantitative analysis. Additionally, to the best of our knowledge, no prior work has integrated DEQ with imbalanced scenarios. In our study, we integrate DEQ with layer-peeled models, add constraints with respect to weights $\boldsymbol{W}_{\text{DEQ}}$, and consider the results of fixed-point iteration as the output of DEQ. Therefore, we analyze the performance of $\mathcal{NC}$ in DEQ by continuously deriving the lower bound of the loss function under certain constraints, allowing us to assess how $\mathcal{NC}$ manifests in the training performance of the network. Similarly, we apply the same operations to explicit neural networks for comparison. Our results show that DEQ performs similarly to explicit neural networks under balanced settings. We further extend the dataset to imbalanced conditions and analyze the $\mathcal{NC}$ performance in DEQ, explaining why DEQ tends to outperform explicit neural networks under mild conditions. We systematically analyze performance in terms of feature convergence, distance to the Simplex ETF, and the parallel relationship between extracted features and classifier weights. These analyses uncover the reasons behind the superior performance of DEQ compared to explicit neural networks during training. Additionally, the experimental results in both balanced and imbalanced scenarios validate our theoretical analyses.

Our main contributions are:

- We systematically analyzed the representation of DEQ from the $\mathcal{NC}$ perspective and compared their performance with explicit neural networks. Our theoretical analysis shows that both DEQ and explicit neural networks exhibit the $\mathcal{NC}$ phenomenon in balanced datasets.

- Under imbalanced settings, we theoretically proved the convergence of extracted features to the vertices of a simplex ETF and alignment with classifier weights under certain conditions, demonstrating DEQ's advantages over explicit neural networks under some mild conditions.

- Experimental results on Cifar-10 and Cifar-100 validated our theoretical findings for distinguishing the differences between DEQ and explicit neural networks.

## 2   Background and related works

We consider a classification task with $K$ classes. Let $n_k$ denote the number of training samples in each class $k$, and $N = \sum_{k=1}^{K} n_k$ represent the total number of training samples. A traditional neural network can be expressed as a mapping:

$$\psi(\boldsymbol{x}) = \boldsymbol{W}\phi(\boldsymbol{x}) + \boldsymbol{b}, \tag{1}$$

where $\phi(\boldsymbol{x}) : \mathbb{R}^{\text{in} \times N} \to \mathbb{R}^{D \times N}$ is the feature extraction, $\boldsymbol{W} \in \mathbb{R}^{K \times D}$ and $\boldsymbol{b} \in \mathbb{R}^{K}$ are the classifiers and bias in the last layer, respectively. For simplicity, we consider the bias-free case and omit the term $\boldsymbol{b}$. Besides, we will denote $\boldsymbol{H} = \phi(\boldsymbol{x})$ in later sections.

### 2.1   Deep Equilibrium Models

There have been numerous neural network architectures designed for various practical tasks from different perspectives [17, 32, 38, 39, 48]. DEQ, a typical implicit network [13, 52], incorporates unrolling methods [12, 41], which are devised for training arbitrarily deep networks by integrating all the network layers into one [3, 4, 5, 35, 36, 58].

Let $f_\theta(\boldsymbol{z}, \boldsymbol{x})$ represent a DEQ layer with input $\boldsymbol{x}$ parameterized by $\theta$. When $\boldsymbol{z}^\star$ reaches the equilibrium point, it satisfies:

$$g_\theta(\boldsymbol{z}^\star, \boldsymbol{x}) \triangleq f_\theta(\boldsymbol{z}^\star, \boldsymbol{x}) - \boldsymbol{z}^\star = 0. \tag{2}$$

The forward procedure mostly employs the Broyden solver [7] for iterative solving:

$$\boldsymbol{z}_{t+1} = \boldsymbol{z}_t - \boldsymbol{B}_t^{-1} g_\theta(\boldsymbol{z}_t, \boldsymbol{x}), \tag{3}$$

where $\boldsymbol{B}_t^{-1}$ refers to the approximation of inverse matrix $\nabla_{\boldsymbol{z}}^{-1} g_\theta(\boldsymbol{z}_t, \boldsymbol{x})$, as well as the same parameter $\theta$ shared across iterations. However, the solution can be quite unstable, and efforts have been made to enhance stability and robustness [34, 44, 55, 56]. Especially, regarding the computation of the inverse matrix, it can be expanded in the form of a Neumann series [18, 60]. Besides, accelerating and stabilizing the backward procedure is also an important issue in DEQ [15].

## 2.2 Neural Collapse $\mathcal{NC}$

The phenomenon of $\mathcal{NC}$ was initially uncovered by [42], which is considered an intriguing regularity in neural network training with many elegant geometric properties [50, 61, 66]. When the model is at the terminal phase of training (TPT), or more precisely, achieves zero training error, the within-class means of features and the classifier vectors converge to the vertices of a simplex Equiangular Tight Frame (ETF) on a balanced dataset.

**Definition 2.1.** (Simplex Equiangular Tight Frame) A collection of points $\boldsymbol{s}_i \in \mathbb{R}^D$, $i = 1, 2, \cdots, K$, is said to be a simplex equiangular tight frame if

$$\boldsymbol{S} = \alpha \sqrt{\frac{K}{K-1}} \boldsymbol{P}\left(\boldsymbol{I}_K - \frac{1}{K} \mathbf{1}_K \mathbf{1}_K^T\right), \tag{4}$$

where $\alpha$ is a non-zero scalar, $\boldsymbol{S} = [\boldsymbol{s}_1, \cdots, \boldsymbol{s}_k] \in \mathbb{R}^{D \times K}$, $\boldsymbol{I}_K \in \mathbb{R}^{K \times K}$ is the identity matrix, $\mathbf{1}_K$ is the ones vector, and $\boldsymbol{P} \in \mathbb{R}^{D \times K} (D \geq K)$ is a partial orthogonal matrix such that $\boldsymbol{P}^T \boldsymbol{P} = \boldsymbol{I}_K$.

$\mathcal{NC}$ incorporates the following four properties of the last-layer features and classifiers in deep learning training on balanced datasets:

$\mathcal{NC}1$: **Variability collapse:** The feature within-class converges to a unique vector, *i.e.*, for any sample $i$ in the same class $k$, its feature $\boldsymbol{h}_{k,i}$ satisfies $\|\boldsymbol{h}_{k,i} - \bar{\boldsymbol{h}}_k\| \to 0, k \in [k]$, with the training procedure.

$\mathcal{NC}2$: **Convergence to simplex ETF:** The mean value $\boldsymbol{h}^\star$ of optimal features for each class collapses to the vertices of the simplex ETF.

$\mathcal{NC}3$: **Convergence to self-duality:** The class means and the classifier weights mutually converge: $\frac{\boldsymbol{W}^\star}{\|\boldsymbol{W}\|} = \frac{\boldsymbol{H}^\star}{\|\boldsymbol{H}\|}$.

$\mathcal{NC}4$: **Nearest Neighbor:** The classifier determines the class based on the Euclidean distances among the feature vector and the classifier weights.

## 2.3 Layer-peeled model under balanced and imbalanced conditions

Current studies often focus on the case where only the last-layer features and classifier are learnable without considering the layers in the backbone network under the assumption of Unconstrained Features Mode (UFM) [66], which can also be referred to as the Layer-peeled Model [14, 28]. First, we define the feasible set of parameters:

$$\mathcal{C} = \left\{ \boldsymbol{w}_k, h_{k,i} \mid \frac{1}{K} \sum_{k=1}^K \|\boldsymbol{w}_k\|^2 \leq E_W, \frac{1}{K} \sum_{k=1}^K \frac{1}{n_k} \sum_{i=1}^{n_k} \|\boldsymbol{h}_{k,i}\|^2 \leq E_H \right\}. \tag{5}$$

**Definition 2.2.** (Layer-peeled Model) When $\boldsymbol{H}$ and $\boldsymbol{W}$ are the last layer classifier and weights respectively, then the optimization process of the neural network can be reformulated as:

$$\min_{\boldsymbol{W}, \boldsymbol{H}} \quad \frac{1}{N} \sum_{k=1}^K \sum_{i=1}^{n_k} \mathcal{L}(\boldsymbol{W} \boldsymbol{h}_{k,i}, \boldsymbol{y}_k) \quad \text{s.t. } \boldsymbol{w}_k, \boldsymbol{h}_{k,i} \in \mathcal{C}, \tag{6}$$

where $E_H$ and $E_W$ are two predefined values, $N$ refers to the total number of samples.

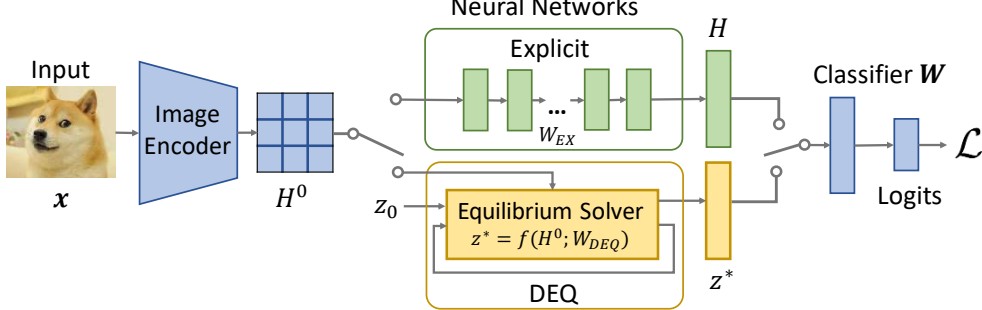

Figure 1: Illustration of feature extraction. After extracting feature maps $\boldsymbol{H}^0$, further features $\boldsymbol{H}$ or $\boldsymbol{z}^\star$ can be obtained by passing through an explicit neural network or DEQ. The final step involves the classifier to obtain predicted logits. To ensure a fair comparison, we standardize the backbone network and its output $\boldsymbol{H}^0$ across all conditions.

It should be noted that all the loss functions $\mathcal{L}$ analyzed in our study are cross-entropy, as most current research focuses on this widely used deep learning classification loss function [23, 31]. And though the optimization program is nonconvex; however, it can generally be mathematically tractable for analysis. Besides, experiments with unregularized loss function and randomly initialized gradient descent typically converge to non-collapse global minimizers [51].

Under UFM, most $\mathcal{NC}$ studies are based on 1-2 conventional layers of weights, However, there is also work [10, 51] that extends it to analyze $M$ linear layers. Additionally, various studies have revealed additional characteristics of $\mathcal{NC}$, such as its impact on generalization [16, 25, 27, 61], its influence on feature learning [45], global optimality of the network [64, 66] and others. Therefore, $\mathcal{NC}$ is a very efficient tool to analyze the performance of neural networks.

**Imbalanced learning** However, $\mathcal{NC}$ will not occur under imbalanced settings generally. This phenomenon arises due to the imbalance in sample quantities, leading to challenges in adequately fitting features for certain classes. This is commonly referred to as *minority collapse* [8, 14]. As the degree of imbalance increases, it is expected that classifiers for minority classes converge. When Minority Collapse occurs, the neural network predicts equal probabilities for all minority classes, regardless of the input.

To enhance learning performance in imbalanced scenarios [62] and mitigate the effects of minority collapse, several methods have been proposed. [14] introduced convex relaxation, modifying a loss function [57], and incorporating a regularization term [37]. The reweighted approach is also widely applied, with some studies measuring it based on sample quantities [47, 59]. Additionally, adaptive techniques such as AutoBalance [33] have been introduced, which incorporates a bilevel optimization framework, along with logit balance [46, 54, 63, 65].

## 3   Comparison under balanced setting

In this section, we first analyze the $\mathcal{NC}$ phenomenon in DEQ under balanced settings. As illustrated in Figure 1, after completing the initial feature extraction, we further examine the feature $\boldsymbol{H}$ obtained respectively by explicit neural networks and DEQ to reveal the $\mathcal{NC}$ phenomenon.

### 3.1   $\mathcal{NC}$ in Explicit Neural Networks

Building upon (6), we analyze $\mathcal{NC}$ in explicit neural networks by considering the following constrained optimization problem during training:

$$\min_{\boldsymbol{W}, \boldsymbol{W}_{\mathrm{EX}}, \boldsymbol{H}^0} \frac{1}{N} \sum_{k=1}^{K} \sum_{i=1}^{n} \mathcal{L}(\boldsymbol{W}\boldsymbol{W}_{\mathrm{EX}}\boldsymbol{h}_{k,i}^0, \boldsymbol{y}_k) \tag{7}$$

$$\text{s.t.} \quad \|\boldsymbol{W}_{\mathrm{EX}}\|_F \leq E_H; \boldsymbol{w}_k, \boldsymbol{h}_{k,i} \in \mathcal{C},$$

where each $n_k$ is set to $n$ under the balanced setting, $\boldsymbol{W}_{\text{EX}}$ represents the subsequent network weights. For ease of comparison with DEQ, we assume that the final feature is represented as $\boldsymbol{H} = \boldsymbol{W}_{\text{EX}}\boldsymbol{H}^0$. Traditional neural network structures are nonconvex, making them challenging to analyze due to their highly interactive nature. Employing the layer-peeled model alleviates the difficulty of $\mathcal{NC}$ analysis.

## 3.2 $\mathcal{NC}$ in Deep Equilibrium models

Building upon recent investigations into the $\mathcal{NC}$ phenomenon, we embrace the layer-peeled model, where the last-layer features $\boldsymbol{h} = \phi(\boldsymbol{x})$ (equilibrium points in DEQ $\boldsymbol{z}^\star$) as unconstrained optimization variables. Accordingly, we add the following constraints to enforce $\mathcal{NC}$ in DEQ:

$$\mathcal{C}_{\text{DEQ}} \triangleq \left\{ \boldsymbol{z}^\star, \boldsymbol{W}_{\text{DEQ}} \middle| \boldsymbol{z}^\star = f(\boldsymbol{H}^0; \boldsymbol{W}_{\text{DEQ}}), \ \|\boldsymbol{W}_{\text{DEQ}}\|_F \leq E_H \right\}. \tag{8}$$

Compared to explicit layers, the active parameter in Deep Equilibrium models is $W_{\text{DEQ}}$, hence imposing restrictions on it to align with the same feasible space. Then the formulation of DEQ with $\mathcal{NC}$ becomes:

$$\min_{\boldsymbol{W}, \boldsymbol{W}_{\text{DEQ}}, \boldsymbol{z}^\star, \boldsymbol{H}^0} \quad \frac{1}{N} \sum_{k=1}^{K} \sum_{i=1}^{n_k} \mathcal{L}(\boldsymbol{W}\boldsymbol{z}^\star, \boldsymbol{y}_k) \tag{9}$$
$$\text{s.t.} \quad \boldsymbol{w}_k, \boldsymbol{h}_{k,i} \in \mathcal{C}; \boldsymbol{z}^\star, \boldsymbol{W}_{\text{DEQ}} \in \mathcal{C}_{\text{DEQ}}.$$

No matter whether under DEQ or explicit neural networks, these constraints must be imposed. This is because when these constraints are satisfied and the loss function reaches its lower bound, the $\mathcal{NC}$ phenomenon is guaranteed. In our theoretical analysis, we assume that the DEQ is linear, that is, $\boldsymbol{z}^\star = \text{fixed-point}(f_\theta(\boldsymbol{x}), \boldsymbol{z}) = \sum_{i=0}^{\infty} \boldsymbol{W}_{\text{DEQ}}^i \boldsymbol{x}$. Detailed analysis incorporating these constraints is provided in Appendix B.

The following theorem elucidates the specific scenarios in which the $\mathcal{NC}$ phenomenon occurs. For a fair comparison, we assume that the extracted features $\boldsymbol{H}^0$ of the image encoder are the same in the derivation.

**Theorem 3.1.** *(Feature collapse of explicit fully connected layers and implicit deep equilibrium models under balanced setting) Suppose (7) and (9) reaches its minimal, then*

$\mathcal{NC}1$: *For $\forall \ k = 1, 2, \cdots, K$ and $\forall \ i = 1, 2, \cdots, n$:*

$$\boldsymbol{W}_{EX}\boldsymbol{h}_{k,i}^0 = \boldsymbol{W}_{EX}\boldsymbol{h}_k^0,$$

*where $\boldsymbol{h}_k^0 = \sum_{i \in \tau(k)} \boldsymbol{h}_{k,i}^0$. Similarly, if the model is DEQ, then*

$$f(\boldsymbol{h}_{k,i}^0; \boldsymbol{W}_{DEQ}) = f(\boldsymbol{h}_k^0; \boldsymbol{W}_{DEQ}).$$

$\mathcal{NC}2$: *The classifier aligns to the Simplex ETF, regardless of whether explicit neural network and DEQ are applied:*

$$\boldsymbol{W}\boldsymbol{W}^T = \sqrt{E_W/E_H}\boldsymbol{W}\boldsymbol{W}_{EX}\boldsymbol{H}^0$$
$$= \sqrt{E_W/E_H}\boldsymbol{W}f(\boldsymbol{H}^0; \boldsymbol{W}_{DEQ})$$
$$= \frac{KE_W}{K-1}\left(\mathbf{1}_K - \frac{1}{K}\mathbf{1}_K\mathbf{1}_K^T\right).$$

$\mathcal{NC}3$: *For $\forall \ k = 1, 2, \cdots, K$, the feature aligns to the weights:*

$$\boldsymbol{W}_{EX}\boldsymbol{h}_k^0 \propto \boldsymbol{W}_k.$$

*In DEQ cases:*

$$f(\boldsymbol{h}_k^0; \boldsymbol{W}_{DEQ}) \propto \boldsymbol{W}_k.$$

The theorem demonstrates that when the network training reaches its limit, i.e., when the loss function reaches its minimum, the $\mathcal{NC}$ phenomenon emerges regardless of whether the chosen network is DEQ or explicit neural network. Besides, in certain scenarios, the lower bound of the loss function for DEQ is relatively smaller compared to explicit neural networks. More detailed proofs are in Appendix Section B.

# 4 Comparison under imbalanced setting

In this section, we analyze the performance differences between DEQ and explicit neural network on imbalanced datasets. We observe that, unlike in balanced scenarios, as long as certain conditions are met, the advantages of DEQ over explicit neural network become more pronounced on imbalanced datasets. And we provide theoretical evidence to support this phenomenon.

Suppose the total number of classes is $K$, with $K_A$ being the number of majority classes and $K_B = K - K_A$ being the number of minority classes. Each majority class has $n_A$ samples, and each minority class has $n_B$ samples. The total number of samples is given by $N = K_A n_A + K_B n_B$. Note that $n_A > n_B$ with no requirement for $K_A$ to be greater than $K_B$. We first start with the loss function, which can be partitioned into two components as follows:

$$\min_{\boldsymbol{W}, \tilde{\boldsymbol{W}}, \boldsymbol{H}^0} \frac{K_A n_A}{N} \sum_{k=1}^{K_A} \sum_{i=1}^{n_A} \mathcal{L}(\boldsymbol{W}\tilde{\boldsymbol{W}}\boldsymbol{H}^0, \boldsymbol{y}_k) + \frac{K_B n_B}{N} \sum_{k=K_A+1}^{K_B} \sum_{i=1}^{n_B} \mathcal{L}(\boldsymbol{W}\tilde{\boldsymbol{W}}\boldsymbol{H}^0, \boldsymbol{y}_k), \quad (10)$$

$$\text{s.t.} \quad \tilde{\boldsymbol{W}} \in \{\mathcal{C}_{\text{EX}} \text{ or } \mathcal{C}_{\text{DEQ}}\}, \quad \boldsymbol{w}_k, \boldsymbol{h}_{k,i} \in \mathcal{C},$$

where $\tilde{\boldsymbol{W}}$ represents the weights of Deep Equilibrium Models $\boldsymbol{W}_{\text{DEQ}}$ and explicit neural network $\boldsymbol{W}_{\text{EX}}$. To analyze the $\mathcal{NC}$ phenomenon, we present the results in the following theorem:

**Theorem 4.1.** *(Neural Collapse under imbalanced settings on explicit neural networks and deep equilibrium models)*

*When the loss function reaches the minimum, then*

$\mathcal{NC}1$: *For $\forall\, k = 1, 2, \cdots, K$ and $\forall\, i = 1, 2, \cdots, n$:*

$$\boldsymbol{W}_{EX} \boldsymbol{h}_{k,i}^0 = \boldsymbol{W}_{EX} \boldsymbol{h}_k^0,$$

*where $\boldsymbol{h}_k^0 = \sum_{i \in \tau(i)} \boldsymbol{h}_{k,i}^0$. Similarly, if the model is DEQ, then*

$$f(\boldsymbol{h}_{k,i}^0; \boldsymbol{W}_{DEQ}) = f(\boldsymbol{h}_k^0; \boldsymbol{W}_{DEQ}).$$

$\mathcal{NC}2$: *Not exists, but the results of explicit neural network and DEQ can be compared:*

*Here we denote $\left(\boldsymbol{h}_k^0\right)^T \boldsymbol{h}_{k'}^0 = \boldsymbol{m}_{k,k'}$ and $\boldsymbol{S}$ is a $K$-Simplex ETF, if*

$$E_H < 2\boldsymbol{S}_{ij} - \boldsymbol{m}_{ij} < \frac{1}{1 - E_H}$$

*is satisfied, the following inequality*

$$\left\| \left(\boldsymbol{W}_{EX} \boldsymbol{H}^0\right)^T \left(\boldsymbol{W}_{EX} \boldsymbol{H}^0\right) - \boldsymbol{S} \right\|_F > \left\| f^T(\boldsymbol{H}^0; \boldsymbol{W}_{DEQ}) f(\boldsymbol{H}^0; \boldsymbol{W}_{DEQ}) - \boldsymbol{S} \right\|_F$$

*holds.*

$\mathcal{NC}3$: *Similarly as $\mathcal{NC}2$, though it does not exist, the results can still be compared, when*

$$\frac{E_H}{E_w + E_H} + E_H(1 - E_H) < 2$$

*is satisfied, then the cosine distance satisfies:*

$$\cos\left(f(\boldsymbol{h}_k; \boldsymbol{W}_{DEQ}), \boldsymbol{w}_k\right) / \cos\left(\boldsymbol{W}_{EX} \boldsymbol{h}_k, \boldsymbol{w}_k\right) > 1.$$

The detailed proof is in Appendix Section C.

Besides, the conclusion regarding the loss function is quite similar to that of Theorem B.3 under balanced settings. As analyzed in (43) and (44) in the Appendix, the lower bound of the loss function in DEQ is still lower than that in explicit neural network, where the performance of learned features is more evident in Figure 2, where we use t-SNE [53] and Gram matrix of features to describe the performance of two models. Although the phenomenon of $\mathcal{NC}2$ and $\mathcal{NC}3$ does not exist, we have discovered in Theorem 4.1 that under mild conditions, DEQ is superior in terms of $\mathcal{NC}$ compared to

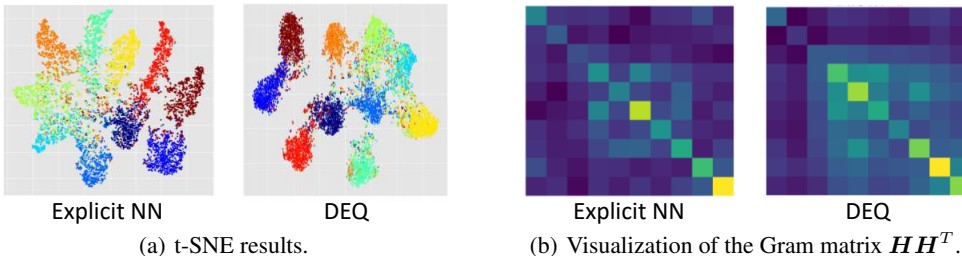

(a) t-SNE results.        (b) Visualization of the Gram matrix $\boldsymbol{HH}^T$.

Figure 2: Under the imbalanced setting for CIFAR-10 with $K_A = 3$ and $R = 10$, the disparity in the learned features between Explicit Neural Networks (left) and DEQ (right).

explicit neural network. Notably, the conditions are easy to satisfy since $E_H$ is generally very small in practice.

A crucial insight is that since DEQ undergoes multiple rounds of parameter adjustments for learning, it can be viewed as having an infinite number of layers, thus possessing greater representational capacity. As the network deepens, the iterative process of forward fixed-point may not necessarily reach the lowest threshold. Therefore, DEQ exhibits a certain degree of generalization for features in the minority class. Given the substantial feature differences among classes under an imbalanced dataset, the learned features by DEQ may demonstrate better adaptability to unseen categories. Consequently, compared to explicit neural network, DEQ tends to enhance performance.

Besides, due to the repeated iterations in solving the fixed-point iteration for some samples in the minority class with a small sample size, the model somewhat engages in multiple learning iterations for the features of samples in this class. This mitigates the impact of imbalanced samples to some extent. However, despite some improvements compared to the explicit neural network, DEQ still faces the issue of minority collapse. This conclusion is further validated in our subsequent experiments. Besides, to further discuss the situation of the dataset in terms of the degree of imbalance, we derived the following proposition:

**Proposition 4.2.** *Denote $R = K_A n_A / N$. When the number of samples in the majority class becomes extremely large, i.e., $R \to 1$, the features of the two kinds of classes will become:*

*Majority classes:*

$$\boldsymbol{W}_{EX}\boldsymbol{h}_{k,i}^0 = \boldsymbol{W}_{EX}\boldsymbol{h}_k^0,$$
$$f(\boldsymbol{h}_{k,i}^0; \boldsymbol{W}_{DEQ}) = f(\boldsymbol{h}_k^0; \boldsymbol{W}_{DEQ}),$$

*where $1 \le k \le K_A$ and $i \in \pi(k)$. Each feature collapses to $K_A$-Simplex ETF.*

*Minority classes:*

$$\boldsymbol{w}_k = \boldsymbol{0},$$
$$\boldsymbol{W}_{EX}\boldsymbol{h}_{k,i}^0 = f(\boldsymbol{h}_{k,i}^0; \boldsymbol{W}_{DEQ}) = \boldsymbol{0},$$

*where $K_A + 1 \le k \le K$ and $i \in \pi(k)$.*

*Here, $\pi(k)$ refers to the samples that belong to the class $k$.*

This situation is equivalent to having a balanced dataset in the majority class, while the minority class, due to its extremely small sample size, contributes almost nothing. In such an extreme scenario, the $\mathcal{NC}$ performance of DEQ and the fully connected layer is nearly indistinguishable similar to Theorem 3.1. Both collapse on majority classes, resulting in a lack of learning features from minority classes meeting the results of (51) and (52). This aligns with the findings in [14], where they provide more specific bounds on the ratio $K_A/K_B$ in their Theorem 5.

## 5   Experiments

In this section, we empirically conducted experiments to validate the correctness of the proposed theorems. Initially, we implemented DEQ on a balanced dataset and compared its $\mathcal{NC}$ performance

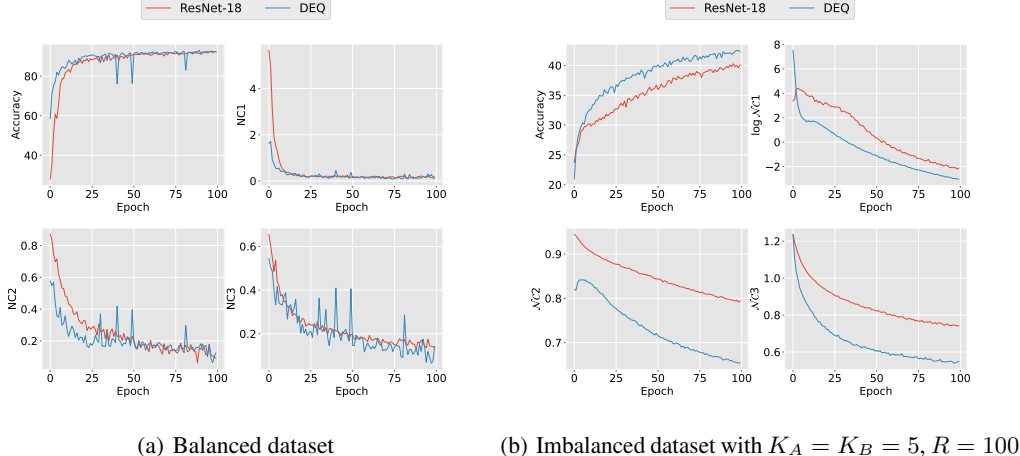

(a) Balanced dataset

(b) Imbalanced dataset with $K_A = K_B = 5, R = 100$

Figure 3: Comparison of accuracy and $\mathcal{NC}$ phenomenon in training Cifar-10 dataset

with that of ResNet. Subsequently, for imbalanced datasets, we tested varying degrees of imbalance by manipulating the quantities of $n_A$ and $n_B$, as well as $K_A$ and $K_B$. The experimental results showed that, on imbalanced datasets, DEQ outperformed Explicit Neural Networks. This finding is consistent with the results reported in [4]. All experiments were implemented using PyTorch on NVIDIA Tesla A40 48GB.

## 5.1 Experiment setup

Without loss of generality, since any traditional neural network can be formulated as a DEQ, we use ResNet18 [22] as the backbone architecture here. As discussed earlier, to utilize the fixed point $z^\star$ learned by DEQ as the extracted feature, we formulate the last ResNet block into a DEQ format, while maintaining the remaining structure identical to ResNet. As mentioned in [5], training with DEQ can lead to instability issues. This is especially noticeable as training progresses, where some samples struggle to converge to a fixed point. To address this, in accordance with their setting, we implement the solver with a threshold $\epsilon$ set to $10^{-3}$ and introduce an early stopping mechanism. If convergence is not achieved within $T > 20$ iterations, we terminate the fixed-point iteration. Additionally, when facing problematic samples during fixed-point solving, we skip them to ensure training stability. During training, we set the learning rate to $1 \times 10^{-4}$ and utilize stochastic gradient descent with a momentum of $0.9$ and weight decay of $5 \times 10^{-4}$. Both $E_W$ and $E_H$ are set to $0.01$. The training phase for each network consists of 100 epochs, with a batch size of 128. In this context, accuracy is assessed by averaging the results from the last 10 epochs and computing their standard deviation.

## 5.2 Performance under balanced conditions

By using the settings in (7) and (9), we compared the performance of DEQ and Explicit NN on Cifar-10 [30] and Cifar-100 [29] for validation, as shown in Figure 3(a). Their $\mathcal{NC}$ performances remain comparable, i.e., DEQ achieves results similar to Explicit NN, corroborating the findings of Theorem 3.1. As for accuracy, from the

Table 1: Comparison of accuracy under balanced settings of Cifar-10 and Cifar-100

|  | Cifar-10 | Cifar-100 |
|---|---|---|
| Explicit NN | $93.05 \pm 0.17$ | $64.35 \pm 0.20$ |
| DEQ | $93.23 \pm 0.13$ | $64.77 \pm 0.36$ |

results in the first column of Table 1, it can be observed that DEQ's accuracy is higher than that of the explicit layer, which aligns with Theorem B.3. However, the increase is only marginal due to the fact that the coefficients $E_H$ and $E_W$ act as scaling factors. Therefore, compared to explicit neural network, DEQ finds it challenging to achieve a significantly lower loss and, consequently, a substantial improvement. Moreover, Explicit NN performs well in fitting balanced datasets, so the accuracy of DEQ does not experience a significant boost in this context.

Here, we manually set the number of epochs to 100 to avoid potential instability issues with DEQ as training deepens. This is because DEQ can be challenging to reach the TPT (Terminal Phase of Training). As the number of parameters increases, achieving fixed-point convergence becomes more difficult, and even parameter explosion may occur. Under the current vanilla design, it is challenging to avoid such instability. Therefore, for a fair comparison, we apply the same training settings to both the implicit DEQ and the explicit neural network. The results in Figure 5 indicate that the test performance at 100 epochs is not significantly different from that at TPT. Since DEQ shares the same backbone as the corresponding explicit neural network, it can still demonstrate better $\mathcal{NC}$ performance after reaching TPT in these cases.

Table 2: Test Accuracy on Cifar-10 and Cifar-100 Dataset with $K_A = 3$

| | | Cifar-10 | | | Cifar-100 | | |
|---|---|---|---|---|---|---|---|
| | $R$ | 10 | 50 | 100 | 10 | 50 | 100 |
| | overall | 72.57±0.25 | 44.32±0.23 | 32.14±0.81 | 41.41±0.56 | 28.18±0.42 | 23.43±0.92 |
| Explicit NN | majority | 96.40±0.32 | 96.80±0.29 | 91.67±0.61 | 73.03±0.62 | 74.53±0.55 | 73.46±0.56 |
| | minority | 62.36±0.12 | 21.83±0.20 | 6.64±0.99 | 27.86±0.39 | 8.31±0.38 | 1.99±1.06 |
| | overall | 73.84±0.72 | 46.08±1.06 | 34.18±1.28 | 43.72±0.60 | 30.46±1.27 | 24.78±1.93 |
| DEQ | majority | 96.68±0.87 | 96.63±0.98 | 93.33±1.36 | 74.16±0.82 | 73.63±0.95 | 74.89±0.88 |
| | minority | 64.06±0.66 | 24.42±1.32 | 8.83±1.08 | 30.67±0.53 | 11.96±1.66 | 3.31±2.45 |

## 5.3 Performance under imbalanced conditions

We conducted experiments with varying configurations with different numbers of majority and minority classes and imbalance degrees. Assume the numbers of majority and minority classes are $(K_A, K_B)$ with corresponding sample sizes $(n_A, n_B)$, the imbalance degree is denoted as $R = n_A/n_B$.We considered different setups for majority and minority class quantities, such as $(3, 7)$, $(5, 5)$, and $(7, 3)$. Additionally, we varied the ratio of sample quantities $R$ between majority and minority classes with values of 10, 50 and 100. We also tested the phenomenon of $\mathcal{NC}$ and accuracy on the Cifar-10 and Cifar-100 datasets, which own a total of 5000 images for each class. Specifically, when $R = 100$ and $(K_A, K_B) = (3, 7)$ for Cifar-10, the number of samples for all classes is $(5000, 5000, 5000, 50, 50, 50, 50, 50, 50, 50)$.

The results for $(K_A, K_B) = (3, 7)$ are shown in Table 2, where the test dataset owns the same distribution as the training dataset. We use "overall", "majority", and "minority" to represent the results across all categories, the majority class, and the minority class, respectively. We contrasted the difference in the training outcomes between the Explicit Neural Network and DEQ, and the superior performance of DEQ compared to Explicit Neural Network confirms DEQ's higher learning potential. This suggests that DEQ can achieve a lower bound on its loss function. The experimental results indicate that DEQ consistently outperforms explicit neural network in accuracy during imbalanced training, aligning with Theorem 4.1. Specifically, we present the outcomes for $(K_A, K_B) = (5, 5)$ with $R = 100$ are depicted in Figure 3(b). The results strongly corroborate Theorem 4.1, affirming DEQ exhibits the same $\mathcal{NC}1$ phenomenon as an explicit neural network under these conditions. However, DEQ outperforms the explicit neural network in terms of $\mathcal{NC}2$ and $\mathcal{NC}3$. Additional experimental results with different parameters are detailed in Appendix Section D.

In addition to the stability considerations discussed in Section 5.2, we refrain from training for an extensive number of epochs due to the imbalance in the samples of the training set. This is because excessive learning rounds might cause the network parameters to predominantly capture information from the majority class, resulting in overfitting its features. This, in turn, diminishes the generalization of learning features from other classes, leading to marginal improvements in accuracy on the test set. As depicted in Figure 3(b), the model has already converged at this point. Moreover, limiting the number of training epochs helps to avoid the gradual instability in the learning process of DEQ.

## 6 Conclusion

In this study, we have systematically analyzed the representation of Deep Equilibrium Models (DEQ) and explicit neural networks under both balanced and imbalanced conditions using the phenomenon of Neural Collapse ($\mathcal{NC}$). Our theoretical analysis demonstrated that $\mathcal{NC}$ is present in DEQ under

balanced conditions. Furthermore, in imbalanced settings, DEQ exhibited notable advantages over explicit neural networks, such as the convergence of extracted features to the vertices of a simplex equiangular tight frame and self-duality properties under mild conditions. These findings highlight the superior performance of DEQ in handling imbalanced datasets. Our experimental results in both balanced and imbalanced scenarios validate the theoretical insights. The current analysis is limited to simple imbalanced scenarios and the linear structure of DEQ models. Future work will expand on this foundation by exploring more general imbalanced scenarios and extending the analysis to more complex forms of DEQ models.

## Acknowledgement

This work was supported by NSFC (No.62303319), Shanghai Sailing Program (22YF1428800), Shanghai Local College Capacity Building Program (23010503100), ShanghaiTech AI4S Initiative SHTAI4S202404, Shanghai Frontiers Science Center of Human-centered Artificial Intelligence (ShangHAI), MoE Key Laboratory of Intelligent Perception and Human-Machine Collaboration (ShanghaiTech University) and Shanghai Engineering Research Center of Intelligent Vision and Imaging.

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

## Appendix Contents

## A    Evaluation metrics of $\mathcal{NC}$

Followed by the settings of [51] and [10], the measurement of $\mathcal{NC}$ are set as follow:

Let $\boldsymbol{h}_k \triangleq \frac{1}{n_k} \sum_{i=1}^{n_k} \boldsymbol{h}_{k,i}$ represent the average of all features within class $k$ and these $K$ classes collectively constitute the average matrix $\bar{\boldsymbol{H}} = [\boldsymbol{h}_1, \cdots, \boldsymbol{h}_K]$. Besides, The global average is defined as $\boldsymbol{h}_G \triangleq \frac{1}{K} \sum_{i=1}^{K} \boldsymbol{h}_k$. Subsequently, the within-class and between-class covariances can be calculated as:

$$
\begin{aligned}
\boldsymbol{\Sigma}_W &\triangleq \frac{1}{N} \sum_{k=1}^{K} \sum_{i=1}^{n_k} (\boldsymbol{h}_{k,i} - \boldsymbol{h}_k)(\boldsymbol{h}_{k,i} - \boldsymbol{h}_k)^T, \\
\boldsymbol{\Sigma}_B &\triangleq \frac{1}{K} \sum_{k=1}^{K} (\boldsymbol{h}_k - \bar{\boldsymbol{h}}_G)(\boldsymbol{h}_k - \bar{\boldsymbol{h}}_G)^T.
\end{aligned}
\tag{11}
$$

$\mathcal{NC}1$ measures the variation of features with-in the same class:

$$
\mathcal{NC}1 = \frac{1}{K} \mathrm{tr} \left( \boldsymbol{\Sigma}_W \boldsymbol{\Sigma}_B^{\dagger} \right),
\tag{12}
$$

where $\boldsymbol{\Sigma}_B^{\dagger}$ denotes the pseudo-inverse of $\boldsymbol{\Sigma}_B$.

$\mathcal{NC}2$ measures similarity between the mean of learned last-layer features $\bar{\boldsymbol{H}}$ and the structure of Simplex ETF:

$$
\mathcal{NC}2 = \left\| \frac{\bar{\boldsymbol{H}}^T \bar{\boldsymbol{H}}}{\|\bar{\boldsymbol{H}}^T \bar{\boldsymbol{H}}\|_F} - \frac{1}{K-1} (\boldsymbol{I}_K - \frac{1}{K} \mathbf{1}_K \mathbf{1}_K^T) \right\|_F.
\tag{13}
$$

$\mathcal{NC}3$ measures similarity of the last-layer feature $\bar{\boldsymbol{H}}$ and weights of classifier $\boldsymbol{W}$:

$$
\mathcal{NC}3 = \left\| \frac{\boldsymbol{W}}{\|\boldsymbol{W}\|_F} - \frac{\bar{\boldsymbol{H}}}{\|\bar{\boldsymbol{H}}\|_F} \right\|.
\tag{14}
$$

Additionally, it is worth noting that all above $\mathcal{NC}$ criteria are exclusively based on the training set. This is because our focus is solely on analyzing learning performance on imbalanced datasets, and generalization is not a primary concern.

# B Proof under balanced setting

## B.1 Problem definition

As different layers in the neural network introduce complexity, the optimization problem is non-convex, and KKT conditions do not guarantee global optimality. Therefore, we consider applying inequality relaxation to the joint optimization problem, obtaining a lower bound for the loss function. By determining the conditions under which the equality holds, we can derive the requirements for the $\mathcal{NC}$ phenomenon. This analysis assumes a balanced setting, where all $\#\tau(k) = n_1 = n_2 = \cdots = n_K = N/K$.

We considered the fully connected layers (explicit) and Deep Equilibrium Models (implicit) under the balanced settings respectively, and then derived the detailed proof.

**(Fully Connected Layers)**

$$\min_{\boldsymbol{W}, \boldsymbol{W}_{\text{EX}}, \boldsymbol{H}} \quad \frac{1}{N} \sum_{k=1}^{K} \sum_{i=1}^{n_k} \mathcal{L}(\boldsymbol{W}\boldsymbol{W}_{\text{EX}}\boldsymbol{h}_{k,i}^0, \boldsymbol{y}_k)$$

$$\text{s.t.} \quad \boldsymbol{h}_{k,i} = \boldsymbol{W}_{\text{EX}}\boldsymbol{h}_{k,i}^0,$$

$$\|\boldsymbol{W}_{\text{EX}}\|_F \leq E_H,$$

$$\frac{1}{K} \sum_{k=1}^{K} \|\boldsymbol{w}_k\|^2 \leq E_W,$$

$$\frac{1}{K} \sum_{k=1}^{K} \frac{1}{n_k} \sum_{i=1}^{n_k} \|\boldsymbol{h}_{k,i}\|^2 \leq E_H,$$

**(Deep Equilibrium Models)**

$$\min_{\boldsymbol{W}, \boldsymbol{W}_{\text{DEQ}}, \boldsymbol{z}_{k,i}^\star} \quad \frac{1}{N} \sum_{k=1}^{K} \sum_{i=1}^{n_k} \mathcal{L}(\boldsymbol{W}\boldsymbol{z}^\star, \boldsymbol{y}_k)$$

$$\text{s.t.} \quad \boldsymbol{z}_{k,i}^\star = f(\boldsymbol{h}_{k,i}^0; \boldsymbol{W}_{\text{DEQ}}),$$

$$\|\boldsymbol{W}_{\text{DEQ}}\|_F \leq E_H,$$

$$\frac{1}{K} \sum_{k=1}^{K} \|\boldsymbol{w}_k\|^2 \leq E_W,$$

$$\frac{1}{K} \sum_{k=1}^{K} \frac{1}{n_k} \sum_{i=1}^{n_k} \|\boldsymbol{z}_{k,i}^\star\|^2 \leq E_H.$$

Note that here $n_1 = n_2 = \cdots = n_k = n$, and $f$ represents the form of Linear DEQ, where we will use $f(\boldsymbol{x}; \boldsymbol{W}_{\text{DEQ}}) = \sum_{i=1}^{\infty} \boldsymbol{W}_{\text{DEQ}}^i \boldsymbol{x}$ for representation in the following proofs.

In a classification task, cross-entropy loss $\mathcal{L}(\boldsymbol{W}\boldsymbol{h}_{k,i}, \boldsymbol{y}_k)$ is regarded as the final loss function. Drawing inspiration from [14], our initial efforts revolve around organizing and simplifying the log function to distinguish the logit in class $k$ from other classes.

First consider the following lemma:

**Lemma B.1.** *Let there be $K$ variables $\delta_1, \delta_2, \cdots, \delta_K$, and the logit of each variable $\delta_k$ satisfies the inequality:*

$$\log\left(\delta_k / \sum_{k=1}^{k} \delta_k\right) \leq M_1 \left(\log \delta_k - \frac{1}{K-1} \sum_{k' \neq k}^{K} \log \delta_k\right) + M_2, \tag{15}$$

*where $M_1$ and $M_2$ are predefined constants.*

*Proof.* Split the sum in the denominator and sequentially introduce weights for each term. Here, define $K$ coefficients such that their sum is 1. Therefore, we have:

$$\frac{C_1}{C_1 + C_2} + \underbrace{C_3 + \cdots + C_3}_{K-1} = 1, \tag{16}$$

that is $C_3 = \dfrac{C_2}{(K-1)(C_1+C_2)}$. Therefore, by Jensen's inequality, we can derive:

$$
\begin{aligned}
\log\left(\delta_k / \sum_{k'=1}^{K}\delta_{k'}\right) &= \log\delta_k - \log\left(\sum_{k'=1}^{K}\delta_{k'}\right) \\
&= \log\delta_k - \log\left(\frac{C_1}{C_1+C_2}\frac{(C_1+C_2)\delta_k}{C_1} + C_3\sum_{k'\neq k}^{K}\frac{\delta_{k'}}{C_3}\right) \\
&\leq \log\delta_k - \frac{C_1}{C_1+C_2}\log\left(\frac{(C_1+C_2)\delta_k}{C_1}\right) - C_3\sum_{k'\neq k}^{K}\log\frac{\delta_{k'}}{C_3} \\
&= M_1\left(\log\delta_k - \frac{1}{K-1}\sum_{k'\neq k}^{K}\log\delta_{k'}\right) + M_2,
\end{aligned}
\tag{17}
$$

where $M_1 = \dfrac{C_2}{C_1+C_2}$ and $M_2 = \dfrac{C_2}{C_1+C_2}\log C_3 - \dfrac{C_1}{C_1+C_2}\log\left(\dfrac{C_1+C_2}{C_1}\right)$. Therefore the lemma is proved. $\qquad\square$

*Remark* B.2. When $C_2/C_1 = \dfrac{1}{K-1}\exp\left(\log\delta_k - \dfrac{1}{K-1}\sum_{k'\neq k}^{K}\log\delta_{k'}\right)$, the right term of the inequality in lemma B.1 reaches its maximum.

*Proof.* Let $\mathcal{M} = M_1\left(\log\delta_k - \dfrac{1}{K-1}\sum_{k'\neq k}^{K}\log\delta_{k'}\right) + M_2$ in Lemma B.1. Then, upon computing the derivatives of $C_1$ and $C_2$, we obtain:

$$
\begin{aligned}
\frac{\partial\mathcal{M}}{\partial C_1} &= \frac{1}{(C_1+C_2)^2}\left(-C_2\frac{\mathcal{M}-M_2}{M_1} + C_2\log\frac{(K-1)C_2}{C_1}\right) \\
\frac{\partial\mathcal{M}}{\partial C_2} &= \frac{1}{(C_1+C_2)^2}\left(C_1\frac{\mathcal{M}-M_2}{M_1} - C_1\log\frac{(K-1)C_2}{C_1}\right).
\end{aligned}
\tag{18}
$$

Combining these two equations yields the conclusion. $\qquad\square$

Next, we substitute the result of each logit into the lemma B.1, from which we can derive:

$$
\begin{aligned}
\mathcal{L} &= -\frac{1}{N}\sum_{k=1}^{K}\sum_{i=1}^{n}\boldsymbol{y}_{k,i}\log\frac{\exp(\boldsymbol{w}_{k,i}^T\boldsymbol{h}_i)}{\sum\limits_{k'=1}^{K}\exp(\boldsymbol{w}_{k',i}^T\boldsymbol{h}_i)} \\
&\geq \frac{C_1}{(C_1+C_2)N(K-1)}\sum_{i=1}^{n}\left[\left(\sum_{k=1}^{K}\boldsymbol{h}_{k,i}\right)^T\left(\sum_{k=1}^{K}\boldsymbol{w}_k\right) - K\sum_{k=1}^{K}\boldsymbol{h}_{k,i}^T\boldsymbol{w}_k\right] + C_4 \\
&= \frac{C_1 K}{(C_1+C_2)N(K-1)}\sum_{k=1}^{K}\sum_{i=1}^{n}(\bar{\boldsymbol{h}}_i - \boldsymbol{h}_{k,i})^T\boldsymbol{w}_k + C_4 \\
&\geq \frac{C_1}{(C_1+C_2)N(K-1)}\left(-\frac{K}{2}\sum_{k=1}^{K}\sum_{i=1}^{n}\|\bar{\boldsymbol{h}}_i - \boldsymbol{h}_{k,i}\|^2/C_5 - \frac{C_5 N}{2}\sum_{k=1}^{K}\|\boldsymbol{w}_k\|^2\right) + C_4,
\end{aligned}
\tag{19}
$$

where the second inequality applies Mean Inequalities, and $C_4 = \dfrac{C_2}{N(C_1+C_2)}\log C_3 - \dfrac{C_1}{N(C_1+C_2)}\log\left(\dfrac{C_1+C_2}{C_1}\right)$. For convenience, we denote $\tilde{\mathcal{L}} = -\dfrac{K}{2}\sum_{k=1}^{K}\sum_{i=1}^{n}\|\bar{\boldsymbol{h}}_i - \boldsymbol{h}_{k,i}\|^2/C_5 -$

$\frac{C_5 N}{2} \sum_{k=1}^{K} \|\boldsymbol{w}_k\|^2$. As the corresponding constraints have already been added in (6), specifically the constraint $\sum_{k=1}^{K} \|\boldsymbol{w}_k\|^2 \leq E_W$, our focus shifts to discussing the situation concerning the first term. Since it represents the features of the final layer, we separately explore the differences in its extraction when using DEQ and fully connected layers. First suppose the extracted feature by the backbone is $\boldsymbol{h}^0$.

## B.2 $\mathcal{NC}$ analysis

We separately discuss the representation of $\mathcal{NC}$ in the cases of Explicit NN and DEQ, and compare the lower bounds of the loss function.

### B.2.1 $\mathcal{NC}$ proof in Explicit neural networks

For convenience, we assume there is only one layer in the feature extractor, that is, $\boldsymbol{h} = \boldsymbol{W}_{\text{EX}} \boldsymbol{h}^0$, then the first term in $\tilde{\mathcal{L}}$ becomes:

$$
\begin{aligned}
-\frac{K}{2} \sum_{k=1}^{K} \sum_{i=1}^{n} \|\bar{\boldsymbol{h}}_i - \boldsymbol{h}_{k,i}\|^2 &= -\frac{K}{2} \sum_{k=1}^{K} \sum_{i=1}^{n} \|\boldsymbol{W}_{\text{EX}}(\bar{\boldsymbol{h}}_i^0 - \boldsymbol{h}_{k,i}^0)\|^2 \\
&\geq -\frac{K}{4} \sum_{k=1}^{K} \sum_{i=1}^{n} \left( \|\boldsymbol{W}_{\text{EX}}\|_F^2 + \|\bar{\boldsymbol{h}}_i^0 - \boldsymbol{h}_{k,i}^0\|^2 \right).
\end{aligned}
\tag{20}
$$

Substituting them into the loss function (19), we can observe that:

$$
\begin{aligned}
\tilde{\mathcal{L}} &\geq -\frac{NK}{4C_5} \|\boldsymbol{W}_{\text{EX}}\|_F^2 - \frac{K}{4C_5} \sum_{k=1}^{K} \sum_{i=1}^{n} \|\bar{\boldsymbol{h}}_i^0 - \boldsymbol{h}_{k,i}^0\|^2 - \frac{C_5 NK}{2} E_W \\
&= -\frac{K^2}{4C_5} \sum_{i=1}^{n} \frac{1}{K} \sum_{k=1}^{K} \left( \|\boldsymbol{h}_{k,i}^0\|^2 - \|\bar{\boldsymbol{h}}_i^0\|^2 \right) - \frac{NK}{4C_5} \|\boldsymbol{W}_{\text{EX}}\|_F^2 - \frac{C_5 NK}{2} E_W \\
&\geq -\frac{KN}{4C_5} E_H - \frac{C_5 KN}{2} E_W + \frac{K^2}{4C_5} \sum_{i=1}^{n} \|\bar{\boldsymbol{h}}_i^0\|^2 - \frac{NK}{4C_5} E_H.
\end{aligned}
\tag{21}
$$

To acquire the lower bound of the loss function, we assign the value $C_5 = \sqrt{E_H / E_W}$, the lower bound becomes:

$$
\inf \mathcal{L}_{\text{EX}} = -\frac{C_1 K}{(C_1 + C_2)(K-1)} \sqrt{E_W E_H} + C_4.
\tag{22}
$$

Furthermore, the condition $\|\bar{\boldsymbol{h}}_i^0\|^2 = 0$ should also be satisfied, indicating that the average of the features for the $i$-th sample, $\frac{1}{K} \sum_{k=1}^{K} \boldsymbol{h}_{k,i}^0$, is equal to zero.

The satisfaction conditions for the inequalities include the following:

- In Eq. (19): The first inequality becomes equality when

$$
\frac{(C_1 + C_2) \boldsymbol{h}_{k,i}^T \boldsymbol{w}_k}{C_1} = \frac{\boldsymbol{h}_{k,i}^T \boldsymbol{w}_{k'}}{C_3},
\tag{23}
$$

that is,

$$
h_{k,i}^T \boldsymbol{w}_k = h_{k,i}^T \boldsymbol{w}_{k'} + \log\left( \frac{C_1(K-1)}{C_2} \right).
\tag{24}
$$

The second inequality is reduced to equality when $\bar{\boldsymbol{h}}_i - \boldsymbol{h}_{k,i} = -C_5 \boldsymbol{w}_k$.

- In Eq. (20): $\|\boldsymbol{W}_{\text{EX}}\|_F^2 = \sum_{k=1}^{K} \sum_{i=1}^{n_k} \|\bar{\boldsymbol{h}}_i^0 - \boldsymbol{h}_{k,i}^0\|^2$.

- In Eq. (21): When the following condition $\frac{1}{K}\sum\limits_{k=1}^{K}\|\boldsymbol{w}_k\|^2 = E_W$ and $\|\boldsymbol{W}_{\mathrm{EX}}\|_F^2 = \frac{1}{K}\sum\limits_{k=1}^{K}\sum\limits_{i=1}^{n_k}\|\boldsymbol{h}_{k,i}^0\|^2 = E_H$ holds, the inequality was reduced to equality.

Since $\|\bar{\boldsymbol{h}}_i^0\|^2 = 0$, it follows that $\|\bar{\boldsymbol{h}}_i\|^2 = \|\boldsymbol{W}_{\mathrm{EX}}\bar{\boldsymbol{h}}_i^0\|^2 = 0$. Combined with the condition $\frac{1}{K}\sum\limits_{k=1}^{K}\|\boldsymbol{w}_k\|^2 = E_W$ and $\frac{1}{K}\sum\limits_{k=1}^{K}\sum\limits_{i=1}^{n_k}\|\boldsymbol{h}_{k,i}\|^2 = E_H$, therefore, $\boldsymbol{h}_k = \boldsymbol{h}_{k,i}$, for $\forall k$, that is, $\mathcal{NC}1$ is proved.

Consequently, $\boldsymbol{h}_{k,i} = C_5\boldsymbol{w}_k$, demonstrating the validity of $\mathcal{NC}3$.

For $\mathcal{NC}2$, since

$$\begin{aligned}
\sqrt{E_H/E_W}\|\boldsymbol{w}_k\|^2 &= \boldsymbol{h}_k\boldsymbol{w}_k = \boldsymbol{h}_k\boldsymbol{w}_{k'} + \log\left(\frac{C_1(K-1)}{C_2}\right) = \boldsymbol{W}_{\mathrm{EX}}\boldsymbol{h}_k^0\boldsymbol{W}_{k'} + \log\left(\frac{C_1(K-1)}{C_2}\right), \\
\sqrt{E_H/E_W}\|\boldsymbol{w}_{k'}\|^2 &= \boldsymbol{h}_{k'}\boldsymbol{w}_{k'} = \boldsymbol{h}_{k'}\boldsymbol{w}_k + \log\left(\frac{C_1(K-1)}{C_2}\right) = \boldsymbol{W}_{\mathrm{EX}}\boldsymbol{h}_{k'}^0\boldsymbol{W}_k + \log\left(\frac{C_1(K-1)}{C_2}\right)
\end{aligned} \tag{25}$$

holds, by the equality conditions, $\|\boldsymbol{w}_k\|^2 = \|\boldsymbol{w}_{k'}\|^2 = E_W$.

Further, $\sum\limits_{k=1}^{K}\boldsymbol{h}_k\boldsymbol{w}_{k'} = \sum\limits_{k=1}^{K}\boldsymbol{W}_{\mathrm{EX}}\boldsymbol{h}_k^0\boldsymbol{w}_{k'} = 0$, as $\boldsymbol{h}_k\boldsymbol{w}_k = \sqrt{E_W E_H}$, so $\boldsymbol{h}_k\boldsymbol{w}_{k'} = -\frac{\sqrt{E_W E_H}}{N-1}$.

Therefore, the $\mathcal{NC}2$ condition satisfies:

$$\boldsymbol{W}\boldsymbol{W}^T = \sqrt{E_W/E_H}\boldsymbol{W}\boldsymbol{H} = \frac{KE_W}{K-1}\left(\mathbf{1}_K - \frac{1}{k}\mathbf{1}_K\mathbf{1}_K^T\right). \tag{26}$$

### B.2.2 $\mathcal{NC}$ proof in DEQ

In the blocks for feature extraction, DEQ can be referred as a mapping from the features by backbone to the output $\boldsymbol{h}^0 \to \boldsymbol{h}^\star$, which can be directly solved using the implicit equation:

$$\boldsymbol{h}^\star = f(\boldsymbol{W}_{\mathrm{DEQ}}; \boldsymbol{h}^0) = \sum_{i=1}^{\infty}\boldsymbol{W}_{\mathrm{DEQ}}^i\boldsymbol{h}^0. \tag{27}$$

Similar as the explicit case, start with the term:

$$-\frac{K}{2}\sum_{k=1}^{K}\sum_{i=1}^{n}\|\bar{\boldsymbol{h}}_i - \boldsymbol{h}_{k,i}\|^2 = \frac{K}{2}\sum_{k=1}^{K}\sum_{i=1}^{n}\left\|\sum_{j=0}^{\infty}\boldsymbol{W}_{\mathrm{DEQ}}^j(\bar{\boldsymbol{h}}_i^0 - \boldsymbol{h}_{k,i}^0)\right\|^2. \tag{28}$$

Since the Neumann series can be regarded as a recursive procedure, denote $\mathcal{G}_{k,i}^j = \sum\limits_{j'=0}^{j}\boldsymbol{W}_{\mathrm{DEQ}}^{j'}(\bar{\boldsymbol{h}}_i^0 - \boldsymbol{h}_{k,i}^0)$ $(j = 0, 1, \cdots, \infty)$, therefore $\mathcal{G}_{k,i}^j = \boldsymbol{W}_{\mathrm{DEQ}}\mathcal{G}_{k,i}^{j-1} + (\bar{\boldsymbol{h}}_i^0 - \boldsymbol{h}_{k,i}^0)$.

$$\begin{aligned}
-\frac{K}{2}\sum_{k=1}^{K}\sum_{i=1}^{n}\left\|\mathcal{G}_{k,i}^j\right\|^2 &= \frac{K}{2}\sum_{k=1}^{K}\sum_{i=1}^{n}\left\|\boldsymbol{W}_{\mathrm{DEQ}}\mathcal{G}_{k,i}^{j-1} + (\bar{\boldsymbol{h}}_i^0 - \boldsymbol{h}_{k,i}^0)\right\|^2 \\
&\geq -\frac{K}{2}\sum_{k=1}^{K}\sum_{i=1}^{n}\left\|\boldsymbol{W}_{\mathrm{DEQ}}\mathcal{G}_{k,i}^{j-1}\right\|^2 - \frac{K}{2}\sum_{k=1}^{K}\sum_{i=1}^{n}\left\|\bar{\boldsymbol{h}}_i^0 - \boldsymbol{h}_{k,i}^0\right\|^2 \\
&\geq -\frac{K}{4}\sum_{k=1}^{K}\sum_{i=1}^{n}\left(\|\boldsymbol{W}_{\mathrm{DEQ}}\|_F^2 + \left\|\mathcal{G}_{k,i}^{j-1}\right\|^2\right) - \frac{K}{2}\sum_{k=1}^{K}\sum_{i=1}^{n}\left\|\bar{\boldsymbol{h}}_i^0 - \boldsymbol{h}_{k,i}^0\right\|_F^2.
\end{aligned} \tag{29}$$

Continuing the recursion, we can obtain:

$$-\frac{1}{2}\|\mathcal{G}_{k,i}^j\|^2 \geq -\left(\frac{1}{2}\right)^{j+1}\|\mathcal{G}_{k,i}^0\|^2 - \left(1 - \frac{1}{2^j}\right)\|\boldsymbol{h}_i^0 - \boldsymbol{h}_{k,i}^0\|^2 - \left(\frac{1}{2} - \frac{1}{2^{j+1}}\right)\|\boldsymbol{W}_{\mathrm{DEQ}}\|_F^2. \tag{30}$$

So, when $j \to \infty$,

$$-\frac{K}{2}\sum_{k=1}^{K}\sum_{i=1}^{n}\left\|\mathcal{G}_{k,i}^{0}\right\|^{2} = \frac{K}{2}\sum_{k=1}^{K}\sum_{i=1}^{n}\left\|\sum_{j=0}^{\infty}\boldsymbol{W}_{\text{DEQ}}^{j}(\bar{\boldsymbol{h}}_{i}^{0}-\boldsymbol{h}_{k,i}^{0})\right\|^{2}$$

$$\geq -K\sum_{k=1}^{K}\sum_{i=1}^{n}\left(\left\|\boldsymbol{h}_{i}^{0}-\boldsymbol{h}_{k,i}^{0}\right\|^{2}-\frac{1}{2}\left\|\boldsymbol{W}_{\text{DEQ}}\right\|_{F}^{2}\right). \tag{31}$$

Therefore, use a similar proof as a fully connected layer,

$$\tilde{\mathcal{L}} \geq -\frac{K}{C_{5}}\sum_{k=1}^{K}\sum_{i=1}^{n}\left\|\boldsymbol{h}_{i}^{0}-\boldsymbol{h}_{k,i}^{0}\right\|^{2}-\frac{NK}{2C_{5}}\left\|\boldsymbol{W}_{\text{DEQ}}\right\|_{F}^{2}-\frac{C_{5}NK}{2}E_{W}$$

$$= -\frac{K}{C_{5}}\sum_{i=1}^{n}\left(\frac{1}{K^{2}}\sum_{k=1}^{K}\left\|\boldsymbol{h}_{k,i}^{0}\right\|^{2}-\left\|\bar{\boldsymbol{h}}_{i}^{0}\right\|^{2}\right)-\frac{NK}{2C_{5}}\left\|\boldsymbol{W}_{\text{DEQ}}\right\|_{F}^{2}-\frac{C_{5}NK}{2}E_{W} \tag{32}$$

$$\geq -\frac{NK}{C_{5}E_{H}}-\frac{C_{5}NK}{2}E_{W}+\frac{K^{2}}{C_{5}}\sum_{i=1}^{n}\left\|\bar{\boldsymbol{h}}_{i}^{0}\right\|^{2}-\frac{NK}{2C_{5}}E_{H}.$$

Set $C_{5}=\sqrt{E_{H}/E_{W}}$, the loss bound of the loss function becomes:

$$\inf \mathcal{L}_{\text{DEQ}} = -\frac{2C_{1}K}{(C_{1}+C_{2})(K-1)}\sqrt{E_{W}E_{H}}+C_{4}. \tag{33}$$

In comparison with the lower bound of the loss function (22), it is evident that the loss function of the DEQ layer is significantly lower than that of the explicit neural network. Since the models are identical, according to Remark B.2, the values of $C_{1}$ and $C_{2}$ are nearly the same. This observation highlights the relatively stronger potential of DEQ compared to Explicit Neural Networks.

Also, the satisfaction conditions for the inequalities in DEQ settings include the following:

- In Eq. (19): The first inequality becomes equality when

$$\frac{(C_{1}+C_{2})\boldsymbol{h}_{k,i}^{T}\boldsymbol{w}_{k}}{C_{1}} = \frac{\boldsymbol{h}_{k,i}^{T}\boldsymbol{w}_{k'}}{C_{3}}, \tag{34}$$

that is,

$$h_{k,i}^{T}\boldsymbol{w}_{k} = h_{k,i}^{T}\boldsymbol{w}_{k'}+\log\left(\frac{C_{1}(K-1)}{C_{2}}\right). \tag{35}$$

The second inequality is reduced to equality when $\bar{\boldsymbol{h}}_{i}-\boldsymbol{h}_{k,i}=-C_{5}\boldsymbol{w}_{k}$. This condition is quite similar to explicit fully connected layers.

- In Eq. (29):
The first inequality:

$$\boldsymbol{W}_{\text{DEQ}}\mathcal{G}_{k,i}^{j-1} = \bar{\boldsymbol{h}}_{i}^{0}-\boldsymbol{h}_{k,i}^{0}, \tag{36}$$

and the second inequality

$$\left\|\boldsymbol{W}_{\text{DEQ}}\right\|_{F}^{2} = \left\|\mathcal{G}_{k,i}^{j-1}\right\|^{2}. \tag{37}$$

- In Eq. (32): When the following condition $\frac{1}{K}\sum_{k=1}^{K}\left\|\boldsymbol{w}_{k}\right\|^{2} = E_{W}$ and $\left\|\boldsymbol{W}_{\text{DEQ}}\right\|^{2} = \frac{1}{K}\sum_{k=1}^{K}\sum_{i=1}^{n_{k}}\left\|\boldsymbol{h}_{k,i}\right\|^{2} = E_{H}$ holds, the inequality was reduced to equality.

To summarize, DEQs are proposed for the memory-saving properties, as the forward passes can leverage any black-box root solvers [3, 5]. However, in terms of forward inference, explicit neural networks have limited learning capacity for data representation since they involve direct expressions computed in a single pass and backward propagation. In contrast, DEQ, lacking a direct explicit form,

requires multiple rounds of parameter adjustments for learning. In each iteration, DEQ introduces input data in a sequential manner, allowing more adjustment space for learning parameters specific to the input. Therefore, to compare the two loss functions, we can derive the following theorem:

**Theorem B.3.** *DEQ achieves a lower bound on the loss function compared to explicit neural network under balanced datasets:*

$$\inf \mathcal{L}_{DEQ} = -2C_1 \frac{K}{K-1} \sqrt{E_W E_H} + C_2,$$

*while the lower bound of loss function of explicit neural network remains:*

$$\inf \mathcal{L}_{EX} = -C_1 \frac{K}{K-1} \sqrt{E_W E_H} + C_2,$$

*where $C_1$ and $C_2$ are two given constants.*

Under the balanced dataset, the sample distribution of each class within each batch is relatively even. Therefore, during the fixed-point iteration process, both DEQ and explicit neural network can learn the features of each class relatively well, without showing significant differences. Besides, from a numerical perspective, the penalties $E_W$ and $E_H$ are generally not set to very large values, especially smaller than 1, so the difference between the two lower bounds in Theorem B.3 may not be substantial. Besides, as analyzed in Remark B.2, we can set $C_2$ in this two equations as identical, and once the propotion of logits in the explicit neural network is greater than the DEQ, the lower bound of loss function in DEQ is lower.

## C  Proof under imbalanced learning

### C.1  Lower bound of the loss function

Consider the loss function:

$$\mathcal{L} = \underbrace{\frac{K_A n_A}{N} \sum_{k=1}^{K_A} \sum_{i=1}^{n_A} \mathcal{L}(\boldsymbol{W}\boldsymbol{h}, \boldsymbol{y}_k)}_{\mathcal{L}_A} + \underbrace{\frac{K_B n_B}{N} \sum_{k=K_A+1}^{K_B} \sum_{i=1}^{n_B} \mathcal{L}(\boldsymbol{W}\boldsymbol{h}, \boldsymbol{y}_k)}_{\mathcal{L}_B}. \tag{38}$$

First analyze the loss in the majority class $\mathcal{L}_A$ and introduce each term in the loss function. Suppose sample $i$ belongs to category $k$, where $1 \leq k \leq K_A$, i.e., $k$ is a majority class.

By applying Jensen's inequalities, we can derive:

$$
\begin{aligned}
& -\log \left( \frac{\exp(\boldsymbol{h}_{k,i}^T \boldsymbol{w}_k)}{\sum_{k'=1}^{K} \exp(\boldsymbol{h}_{k,i}^T \boldsymbol{w}_{k'})} \right) \\
&= -\boldsymbol{h}_{k,i}^T \boldsymbol{w}_k + \log \left( C_1 \exp \left( \frac{\boldsymbol{h}_{k,i}^T \boldsymbol{w}_k}{C_1} \right) + C_2 \sum_{k' \neq k}^{K_A} \exp \left( \frac{\boldsymbol{h}_{k,i}^T \boldsymbol{w}_{k'}}{C_2} \right) + C_3 \sum_{k'=K_A+1}^{K} \exp \left( \frac{\boldsymbol{h}_{k,i}^T \boldsymbol{w}_{k'}}{C_3} \right) \right) \\
&\geq (C_1 - 1)\boldsymbol{h}_{k,i}^T \boldsymbol{w}_k + C_2 \sum_{k' \neq k}^{K_A} \boldsymbol{h}_{k,i}^T \boldsymbol{w}_{k'} + C_3 \sum_{k'=K_A+1}^{K} \boldsymbol{h}_{k,i}^T \boldsymbol{w}_{k'} + const \\
&= C_0 C_4 \left( \frac{1}{K_A} \sum_{k'=1}^{K_A} \boldsymbol{h}_{k,i}^T \boldsymbol{w}_{k'} - \boldsymbol{h}_{k,i}^T \boldsymbol{w}_k \right) + C_0 C_5 \left( \frac{1}{K_B} \sum_{k'=K_A+1}^{K} \boldsymbol{h}_{k,i}^T \boldsymbol{w}_{k'} - \boldsymbol{h}_{k,i}^T \boldsymbol{w}_k \right) + const \\
&= C_0 C_4 \left( \boldsymbol{h}_{k,i}^T \boldsymbol{w}_A - \boldsymbol{h}_{k,i}^T \boldsymbol{w}_k \right) + C_0 C_5 \left( \boldsymbol{h}_{k,i}^T \boldsymbol{w}_B - \boldsymbol{h}_{k,i}^T \boldsymbol{w}_k \right) + const.
\end{aligned}
\tag{39}
$$

Here the value of const is $-C_1 \log C_1 - (k_A - 1)C_2 \log C_2 - K_B C_3 \log C_3$. Besides, $\boldsymbol{w}_A = \frac{1}{K_A} \sum_{k'=1}^{K_A} \boldsymbol{w}_{k'}$ and $\boldsymbol{w}_B = \frac{1}{K_B} \sum_{k'=K_A+1}^{K} \boldsymbol{w}_{k'}$ represent the mean values of the weights in majority and minority classes, respectively.

To ensure the equality conditions hold, suppose there are three adaptive constants $a > 0$, $b > 0$, $c > 0$. Denote $C_1 = \dfrac{a}{a + (K_A - 1)b + K_B c}$, $C_2 = \dfrac{b}{a + (K_A - 1)b + K_B c}$, and $C_3 = \dfrac{c}{a + (K_A - 1)b + K_B c}$. Additionally, to ensure $C_4 + C_5 = 1$, introduce a constant $C_0 = \dfrac{K_A b + K_B c}{a + (K_A - 1)b + K_B c}$, thus $C_4 = \dfrac{K_A b}{K_A b + K_B c}$ and $C_5 = \dfrac{K_B c}{K_A b + K_B c}$.

After aggregating each term in the loss function, we obtain:

$$\frac{1}{K_A n_A} \sum_{k=1}^{K_A} \sum_{i=1}^{n_A} \mathcal{L}(\boldsymbol{W}\boldsymbol{h}, \boldsymbol{y}_k)$$

$$\geq \frac{1}{K_A n_A} \sum_{k=1}^{K_A} \sum_{i=1}^{n_A} C_4 \left( \boldsymbol{h}_{k,i}^T \boldsymbol{w}_A - \boldsymbol{h}_{k,i}^T \boldsymbol{w}_k \right) + C_5 \left( \boldsymbol{h}_{k,i}^T \boldsymbol{w}_B - \boldsymbol{h}_{k,i}^T \boldsymbol{w}_k \right) + const \qquad (40)$$

$$= \frac{1}{K_A} \sum_{k=1}^{K_A} \boldsymbol{h}_k^T (C_4 \boldsymbol{w}_A + C_5 \boldsymbol{w}_B - \boldsymbol{w}_k) + const,$$

where $\boldsymbol{h}_k = \dfrac{1}{n_A} \sum_{i=1}^{n_B} \boldsymbol{h}_{k,i}$.

Subsequently, consider the lower bound of

$$\sum_{k=1}^{K_A} \boldsymbol{h}_k^T (C_4 \boldsymbol{w}_A + C_5 \boldsymbol{w}_B - \boldsymbol{w}_k) \geq -\frac{C_6}{2} \sum_{k=1}^{K_A} \|\boldsymbol{h}_k\|^2 - \sum_{k=1}^{K_A} \frac{1}{2} \|C_4 \boldsymbol{w}_A + C_5 \boldsymbol{w}_B - \boldsymbol{w}_k\|^2 / C_6. \quad (41)$$

Note that this inequality (41) is reduced to equality only when the following equality holds:

$$C_4 \boldsymbol{w}_A + C_5 \boldsymbol{w}_B - \boldsymbol{w}_k = C_6 \boldsymbol{h}_k, \qquad (42)$$

where $1 \leq k \leq K_A$.

Continuing the analysis of inequality (41), the first term on the right-hand side can be bounded as:

**Case 1:** (Explicit fully connected layers)

$$-\sum_{k=1}^{K_A} \|\boldsymbol{h}_k\|^2 = -\sum_{k=1}^{K_A} \|\boldsymbol{W}_{\text{EX}} \boldsymbol{h}_k^0\|^2$$

$$\geq -\frac{1}{2} \left( K_A \|\boldsymbol{W}_{\text{EX}}\|_F + \sum_{k=1}^{K_A} \|\boldsymbol{h}_k^0\|^2 \right) \qquad (43)$$

$$\geq -\frac{1}{2} \left( K_A \|\boldsymbol{W}_{\text{EX}}\|_F + \sum_{k=1}^{K_A} \frac{1}{n_k} \sum_{i=1}^{n_k} \|\boldsymbol{h}_{k,i}^0\|^2 \right)$$

$$\geq -K_A E_H.$$

**Case 2:** (Deep Equilibrium Models)

$$-\sum_{k=1}^{K_A} \|\boldsymbol{h}_k\|^2 = -\sum_{k=1}^{K_A} \left\| (\boldsymbol{I} - \boldsymbol{W}_{\text{DEQ}})^{-1} \boldsymbol{h}_k^0 \right\|^2$$

$$\geq -\frac{1}{2} \left( K_A \sum_{j=0}^{\infty} \|\boldsymbol{W}_{\text{DEQ}}\|_F^j + \sum_{k=1}^{K_A} \|\boldsymbol{h}_k\|^2 \right) \qquad (44)$$

$$\geq -\frac{1}{2} \left( K_A \sum_{j=0}^{\infty} E_H^j + \sum_{k=1}^{K_A} \frac{1}{n_k} \sum_{i=1}^{n_k} \|\boldsymbol{h}_{k,i}\|^2 \right)$$

$$\geq -\frac{1}{2} \left( \frac{1}{1 - E_H} + E_H \right).$$

Compared the lower bound of explicit neural network and DEQ, we can find that:

$$\left(-\sum_{k=1}^{K_A}\|\boldsymbol{h}_k\|^2\right)_{\text{DEQ}} < \left(-\sum_{k=1}^{K_A}\|\boldsymbol{h}_k\|^2\right)_{\text{EX}}$$

for all $E_H \neq 1$.

We now shift our attention to the second term (Ref. Eq [82-83] in [14]):

$$
\begin{aligned}
&-\frac{1}{K_A}\sum_{k=1}^{K_A}\|C_4\boldsymbol{w}_A + C_5\boldsymbol{w}_B - \boldsymbol{w}_k\|^2 \\
&= -\frac{1}{K_A}\sum_{k=1}^{K_A}\|\boldsymbol{w}_k\|^2 + \frac{2}{K_A}\sum_{k=1}^{K_A}\boldsymbol{w}_k^T(C_4\boldsymbol{w}_A + C_5\boldsymbol{w}_B) - \|C_4\boldsymbol{w}_A + C_5\boldsymbol{w}_B\|^2 \\
&= -\frac{1}{K_A}\sum_{k=1}^{K_A}\|\boldsymbol{w}_k\|^2 + 2C_5^2\boldsymbol{w}_A^T\boldsymbol{w}_B + C_4(2-C_4)\|\boldsymbol{w}_A\|^2 - C_5\|\boldsymbol{w}_B\|^2 \\
&= -\frac{1}{K_A}\sum_{k=1}^{K_A}\|\boldsymbol{w}_k\|^2 + \frac{1}{K_A}\sum_{k=K_A+1}^{K}\|\boldsymbol{w}_k\|^2 + C_4(2-C_4)\left\|\boldsymbol{w}_A + \frac{C_5^2}{C_4(2-C_4)}\boldsymbol{w}_B\right\|^2 \qquad (45) \\
&\qquad\qquad\qquad\qquad - \left(C_5^2 + \frac{C_5^2}{C_4(2-C_4)}\right)\|\boldsymbol{w}_B\|^2 \\
&\geq -\frac{K}{K_A}E_W + \left(\frac{1}{K_R} - C_5^2 - \frac{C_5^4}{C_4(2-C_4)}\right)\|\boldsymbol{w}_B\|^2 + \frac{1}{K_A}\sum_{k=K_A+1}^{K}\|\boldsymbol{w}_k - \boldsymbol{w}_B\|^2 \\
&\qquad\qquad\qquad\qquad + C_4(2-C_4)\left\|\boldsymbol{w}_A + \frac{C_5^2}{C_4(2-C_4)}\boldsymbol{w}_B\right\|^2,
\end{aligned}
$$

where $K_R = K_A/K_B$ denotes the ratio of the number of majority classes to minority classes.
In summary, the lower bound of loss function (40) could be simplified as:

$$
\begin{aligned}
\mathcal{L}_A &= \frac{1}{K_A n_A}\sum_{k=1}^{K_A}\sum_{i=1}^{n_A}\mathcal{L}(\boldsymbol{W}\boldsymbol{h}_{k,i}, \boldsymbol{y}_k) \\
&\geq \frac{1}{K_A}\sum_{k=1}^{K_A}\boldsymbol{h}_k^T(C_4\boldsymbol{w}_A + C_5\boldsymbol{w}_B - \boldsymbol{w}_k) + const \\
&\geq -\frac{C_6}{2K_A}\sum_{k=1}^{K_A}\|\boldsymbol{h}_k\|^2 - \frac{1}{2K_A}\sum_{k=1}^{K_A}\|C_4\boldsymbol{w}_A + C_5\boldsymbol{w}_B - \boldsymbol{w}_k\|^2/C_6 + const \qquad (46) \\
&\geq \frac{C_6}{2K_A}M - \frac{KE_W}{2C_6 K_A} + \frac{1}{2C_6}\left(\frac{1}{K_R} - C_5^2 - \frac{C_5^4}{C_4(2-C_4)}\right)\|\boldsymbol{w}_B\|^2 \\
&\qquad + \frac{C_4(2-C_4)}{C_6}\left\|\boldsymbol{w}_A + \frac{C_5^2}{C_4(2-C_4)}\boldsymbol{w}_B\right\|^2 + \frac{1}{2C_6 K_A}\sum_{k=K_A+1}^{K}\|\boldsymbol{w}_k - \boldsymbol{w}_B\|^2 + const,
\end{aligned}
$$

where $M = -K_A E_H$ if the network is a fully connected layer and $M = -\frac{K_A}{2}\left(\frac{1}{1-E_H} + E_H\right)$ if the network is a Deep Equilibrium Model.

Similarly, the loss function w.r.t the minority classes is bounded as:

$$
\begin{aligned}
\mathcal{L}_B &= \frac{1}{K_B n_B} \sum_{k=1}^{K_B} \sum_{i=1}^{n_B} \mathcal{L}(\boldsymbol{W} \boldsymbol{h}_{k,i}, \boldsymbol{y}_k) \\
&= \frac{1}{K_B} \sum_{k=1}^{K_B} \boldsymbol{h}_k^T (C_4 \boldsymbol{w}_A + C_5 \boldsymbol{w}_B - \boldsymbol{w}_k) + const \\
&\geq -\frac{C_6}{2K_B} \sum_{k=1}^{K_B} \|\boldsymbol{h}_k\|^2 - \frac{1}{2K_B} \sum_{k=1}^{K_B} \|C_4 \boldsymbol{w}_A + C_5 \boldsymbol{w}_B - \boldsymbol{w}_k\|^2 / C_6 + const \qquad (47) \\
&\geq \frac{C_6}{2K_B} M - \frac{K E_W}{2 C_6 K_B} + \frac{1}{2 C_6} \left( K_R - C_5^2 - \frac{C_5^4}{C_4(2 - C_4)} \right) \|\boldsymbol{w}_A\|^2 \\
&\quad + \frac{C_5(2 - C_5)}{C_6} \left\| \frac{C_4^2}{C_5(2 - C_5)} \boldsymbol{w}_A + \boldsymbol{w}_B \right\|^2 + \frac{1}{2 C_6 K_B} \sum_{k=1}^{K_A} \|\boldsymbol{w}_k - \boldsymbol{w}_A\|^2 + const.
\end{aligned}
$$

The inequality reduces to equality when the constraints in $\mathcal{C}$ are treated as equalities, achieving the upper bound. Additionally, the following equalities must hold:

$$
C_4 \boldsymbol{w}_A + C_5 \boldsymbol{w}_B - \boldsymbol{w}_k = C_6 \boldsymbol{h}_k, \qquad (48)
$$

where $K_A + 1 \leq k \leq K$.

If $K_R = 1$, i.e., the number of majority classes is equal to the number of minority classes, the results of (46) and (47) are totally equivalent.

Therefore, without loss of generality, assuming $K_A > K_B$, the lower bound of the loss function (38) can be simplified to:

$$
\begin{aligned}
\mathcal{L} &= \mathcal{L}_A + \mathcal{L}_B \\
&\geq \frac{C_6 M}{2} \left( \frac{1}{K_A} + \frac{1}{K_B} \right) + \frac{1}{2 C_6 K_B} \sum_{k=1}^{K_A} \|\boldsymbol{w}_k - \boldsymbol{w}_A\|^2 + \frac{1}{2 C_6 K_A} \sum_{k=1}^{K_B} \|\boldsymbol{w}_k - \boldsymbol{w}_B\|^2 \\
&\quad + \frac{C_4(2 - C_4)}{2 C_6} \left\| \boldsymbol{w}_A + \frac{C_5^2}{C_4(2 - C_4)} \boldsymbol{w}_B \right\|^2 + \frac{C_5(2 - C_5)}{2 C_6} \left\| \frac{C_4^2}{C_5(2 - C_5)} \boldsymbol{w}_A + \boldsymbol{w}_B \right\|^2 \qquad (49) \\
&\quad + \frac{1}{2 C_6} \left( \frac{1}{K_R} - C_5^2 - \frac{C_5^4}{C_4(2 - C_4)} \right) \|\boldsymbol{w}_B\|^2 + \frac{1}{2 C_6} \left( K_R - C_5^2 - \frac{C_5^4}{C_4(2 - C_4)} \right) \|\boldsymbol{w}_A\|^2 + const.
\end{aligned}
$$

## C.2 $\mathcal{NC}$ Analysis

As analyzed in (43) and (44), when it reaches the minimal value, each $\boldsymbol{h}_{k,i} = \boldsymbol{h}_k$ for $\forall k = 1, 2 \cdots, K_A$. Similarly, this holds for minority class with $K_A + 1 \leq k \leq K$. This implies that, in an imbalanced scenario, both DEQ and fully connected layer exhibit feature collapse, i.e., $\mathcal{NC}1$ is still present.

As we need to calculate the lower bound of the loss function, it is essential to minimize the terms $\mathcal{L}_A + \mathcal{L}_B$ as much as possible.

Therefore, consider the gradient with respect to $\boldsymbol{w}_k$ for majority class and $\boldsymbol{w}_k$ for minority class, respectively. First compute the case with $1 \leq k \leq K_A$.

$$
\begin{aligned}
\frac{\partial \mathcal{L}}{\partial \boldsymbol{w}_k} &= \frac{1}{C_6 K_B} \left( 1 - \frac{1}{K_A} \right) (\boldsymbol{w}_k - \boldsymbol{w}_A) + \frac{C_4(2 - C_4)}{C_6 K_A} \left( \boldsymbol{w}_A + \frac{C_5^2}{C_4(2 - C_4)} \boldsymbol{w}_B \right) \\
&\quad + \frac{C_4^2}{K_A C_6} \left( \frac{C_4^2}{C_5(2 - C_5)} \boldsymbol{w}_A + \boldsymbol{w}_B \right) \qquad (50) \\
&\quad + \frac{1}{K_A C_6} \left( K_R - C_5^2 - \frac{C_5^4}{C_4(2 - C_4)} \right) \boldsymbol{w}_A = 0.
\end{aligned}
$$

So, we can derive that

$$\left(K_R - \frac{1}{K_B}\right)\boldsymbol{w}_k + \frac{1}{1 - C_4^2}\boldsymbol{w}_B$$
$$+ \left(\frac{1}{K_B} + C_4(2 - C_4) + \frac{C_4^4}{C_5(2 - C_5)} - C_5^2 - \frac{C_5^2}{C_4(2 - C_4)}\right)\boldsymbol{w}_A = 0. \tag{51}$$

One important note here is that when the proportion of majority class samples approaches infinity, i.e., $C_4 \to 1$, we have $\frac{1}{1-C_4^2} \to 0$. In this scenario, the weights $\boldsymbol{w}_k$ belonging to the majority class are almost exclusively related to $\boldsymbol{w}_A$, and have little dependence on the average of the minority class $\boldsymbol{w}_B$, which validates the results of Proposition 4.2.

Similarly, if $K_A + 1 \le k \le K$, the following equality will hold to ensure optimality of $\boldsymbol{w}_k$ in minority classes of (49):

$$\left(\frac{1}{K_R} - \frac{1}{K_A}\right)\boldsymbol{w}_k + \frac{1}{1 - C_5^2}\boldsymbol{w}_A$$
$$+ \left(\frac{1}{K_A} + C_5(2 - C_5) + \frac{C_5^4}{C_4(2 - C_4)} - C_4^2 - \frac{C_4^2}{C_5(2 - C_5)}\right)\boldsymbol{w}_B = 0. \tag{52}$$

Next, we consider the conditions for the validity in $\mathcal{NC}2$ and $\mathcal{NC}3$, then compare the performance of DEQ and explicit neural network.

Therefore, for the majority class $1 \le k \le K_A$, suppose it reaches its minimum value, recall the condition (42), and combined with (51), we can derive:

$$C_6\boldsymbol{h}_k^T\boldsymbol{h}_{k'} = \left(C_4 + \frac{K_B C_A}{K_A - 1}\right)\boldsymbol{w}_A^T\boldsymbol{h}_{k'} + \left(C_5 + \frac{K_B}{(K_A - 1)(1 - C_4^2)}\right)\boldsymbol{w}_B^T\boldsymbol{h}_{k'}. \tag{53}$$

Similarly, for the minority class $K_A + 1 \le k \le K$, combine (48) with (52), we obtain:

$$C_6\boldsymbol{h}_k^T\boldsymbol{h}_{k'} = \left(C_4 + \frac{K_A}{(K_B - 1)(1 - C_5^2)}\right)\boldsymbol{w}_A^T\boldsymbol{h}_{k'} + \left(C_5 + \frac{K_A C_B}{K_B - 1}\right)\boldsymbol{w}_B^T\boldsymbol{h}_{k'}. \tag{54}$$

In the above two equations, $k' = 1, 2, \cdots, K$. And we denote the coefficient of $\boldsymbol{w}_A$ in Eq. (51) and the coefficient of $\boldsymbol{w}_B$ in Eq. (52) as $C_A$ and $C_B$ respectively for simplicity. After this deviation, we can find that both of the coefficients of $\boldsymbol{w}_A^T\boldsymbol{h}_{k'}$ and $\boldsymbol{w}_B^T\boldsymbol{h}_{k'}$ are constants.

It can be obviously concluded that $\mathcal{NC}2$ and $\mathcal{NC}3$ do not hold under imbalanced dataset conditions. However, we can still compare the numerical differences between them under DEQ and fully connected layer. By adaptively specifying parameters $C_4$ and $C_5$, we can denote $(\boldsymbol{h}_k^0)^T\boldsymbol{h}_{k'}^0 = \boldsymbol{m}_{k,k'}$.

Thus, by considering all the equality conditions in (46), we can measure the distance from the features to the Simplex ETF.

$$C_6\boldsymbol{h}_{k'}^T\boldsymbol{h}_k = C_4\boldsymbol{h}_{k'}^T\boldsymbol{w}_A + C_5\boldsymbol{h}_{k'}^T\boldsymbol{w}_B - \boldsymbol{h}_{k'}^T\boldsymbol{w}_k. \tag{55}$$

**Case 1:** (Explicit fully connected layers)

$$\begin{aligned}
C_6(h_{k'}^0)^T\boldsymbol{h}_k &= C_4\boldsymbol{W}_{\text{EX}}(h_{k'}^0)^T\boldsymbol{w}_A + C_5\boldsymbol{W}_{\text{EX}}(h_{k'}^0)^T\boldsymbol{w}_B - \boldsymbol{W}_{\text{EX}}(h_{k'}^0)^T\boldsymbol{w}_k \\
&= \boldsymbol{W}_{\text{EX}}\left(C_4\boldsymbol{h}_{k'}^T\boldsymbol{w}_A + C_5\boldsymbol{h}_{k'}^T\boldsymbol{w}_B - \boldsymbol{h}_{k'}^T\boldsymbol{w}_k\right) \\
&\le \frac{1}{2}\|\boldsymbol{W}_{\text{EX}}\|_F + \frac{1}{2}\left\|C_4\boldsymbol{h}_{k'}^T\boldsymbol{w}_A + C_5\boldsymbol{h}_{k'}^T\boldsymbol{w}_B - \boldsymbol{h}_{k'}^T\boldsymbol{w}_k\right\| \\
&= E_H + \frac{1}{2}\boldsymbol{M}.
\end{aligned} \tag{56}$$

**Case 2:** (Deep Equilibrium Models)

$$C_6(h_{k'}^0)^T \boldsymbol{h}_k = C_4(\boldsymbol{I} - \boldsymbol{W}_{\text{DEQ}})^{-1}(h_{k'}^0)^T \boldsymbol{w}_A + C_5(\boldsymbol{I} - \boldsymbol{W}_{\text{DEQ}})^{-1}(h_{k'}^0)^T \boldsymbol{w}_B$$
$$- (\boldsymbol{I} - \boldsymbol{W}_{\text{DEQ}})^{-1}(h_{k'}^0)^T \boldsymbol{w}_k$$
$$= (\boldsymbol{I} - \boldsymbol{W}_{\text{DEQ}})^{-1} \left( C_4 \boldsymbol{h}_{k'}^T \boldsymbol{w}_A + C_5 \boldsymbol{h}_{k'}^T \boldsymbol{w}_B - \boldsymbol{h}_{k'}^T \boldsymbol{w}_k \right)$$
$$\leq \frac{1}{2} \| (\boldsymbol{I} - \boldsymbol{W}_{\text{DEQ}})^{-1} \|_F + \frac{1}{2} \left\| C_4 \boldsymbol{h}_{k'}^T \boldsymbol{w}_A + C_5 \boldsymbol{h}_{k'}^T \boldsymbol{w}_B - \boldsymbol{h}_{k'}^T \boldsymbol{w}_k \right\| \tag{57}$$
$$= \frac{1}{2(1 - E_H)} + \frac{1}{2} \boldsymbol{M}.$$

Therefore, to compare these two models, we consider the case when the distance of these two models from the Simplex ETF is minimized. We denote each element in the Simplex ETF as $\boldsymbol{s}_{ij}$ and compare the differences between them. When the distance of DEQ is relatively smaller than that of explicit neural network, we can obtain:

$$\left| \frac{1}{2(1 - E_H)} + \frac{1}{2}\boldsymbol{m} - \boldsymbol{s} \right| < \left| \frac{1}{2}E_H + \frac{1}{2}\boldsymbol{m} - \boldsymbol{s} \right|. \tag{58}$$

For simplicity, we only consider the subscripts of $\boldsymbol{s}$ and $\boldsymbol{m}$, and denote ① $= \frac{1}{2(1-E_H)} + \frac{1}{2}\boldsymbol{m} - \boldsymbol{s}$ and ② $= \frac{1}{2}E_H + \frac{1}{2}\boldsymbol{m} - \boldsymbol{s}$. We then classify and discuss their magnitudes.

- ① $> 0$, ② $> 0$:
  Since $\frac{1}{2(1-E_H)} > E_H$, there is a contradiction! Therefore, it does not hold.

- ① $> 0$, ② $< 0$:
  $\frac{1}{2(1-E_H)} + \frac{1}{2}\boldsymbol{m} - \boldsymbol{s} < 0$, $\frac{1}{2}E_H + \frac{1}{2}\boldsymbol{m} - \boldsymbol{s} > 0$, which means $\frac{1}{2(1-E_H)} < E_H$. And that is a contradiction!

- ① $< 0$, ② $> 0$:
  Since $\frac{1}{2(1-E_H)} + \frac{1}{2}\boldsymbol{m} - \boldsymbol{s} > 0$, $\frac{1}{2}E_H + \frac{1}{2}\boldsymbol{m} - \boldsymbol{s} < 0$, we have

  $$E_H < 2\boldsymbol{s} - \boldsymbol{m} < \frac{1}{1 - E_H}.$$

  Besides, by the inequality (58), we have $\frac{1}{2}E_H + \frac{1}{2(1-E_H)} < 2\boldsymbol{s} - \boldsymbol{m}$. Then find the intersection, we obtain that:

  $$\frac{1}{2}E_H + \frac{1}{2(1 - E_H)} < 2\boldsymbol{s} - \boldsymbol{m} < \frac{1}{1 - E_H}.$$

- ① $< 0$, ② $< 0$:
  Since $\frac{1}{2(1-E_H)} + \frac{1}{2}\boldsymbol{m} - \boldsymbol{s} < 0$ and $\frac{1}{2}E_H + \frac{1}{2}\boldsymbol{m} - \boldsymbol{s} < 0$, it implies that $\frac{1}{2(1-E_H)} > E_H$ always holds.
  Therefore, we only need to ensure that

  $$2\boldsymbol{s} - \boldsymbol{m} > \min \left\{ E_H, \frac{1}{1 - E_H} \right\}.$$

Combining these four cases and finding their intersection, we conclude that when the inequality

$$E_H < 2\boldsymbol{s} - \boldsymbol{m} < \frac{1}{1 - E_H} \tag{59}$$

is satisfied, the performance of DEQ is better than that of explicit neural network.

As for $\mathcal{NC}3$, consider the cosine distance with the feature $\boldsymbol{h}_k$ and $\boldsymbol{w}_k$.

**Case 1:** (Explicit fully connected layers)

$$\cos(\boldsymbol{h}_k, \boldsymbol{w}_k)_{\text{EX}} = \frac{\boldsymbol{h}_k^T \boldsymbol{w}_k}{\|\boldsymbol{w}_k\|\|\boldsymbol{h}_k\|}$$
$$= \frac{\boldsymbol{W}_{\text{EX}}\left(\boldsymbol{h}_k^0\right)^T \boldsymbol{w}_k}{\|\boldsymbol{w}_k\|\|\boldsymbol{W}_{\text{EX}}\boldsymbol{h}_k^0\|}$$
$$\geq \frac{2\boldsymbol{W}_{\text{EX}}\left(\boldsymbol{h}_k^0\right)^T \boldsymbol{w}_k}{\|\boldsymbol{w}_k\|^2 + \frac{1}{2}\|\boldsymbol{W}_{\text{EX}}\|^2 + \frac{1}{2}\|\boldsymbol{h}_k^0\|} \tag{60}$$
$$\geq \frac{2E_H\left(\boldsymbol{h}_k^0\right)^T \boldsymbol{w}_k}{E_W + E_H}.$$

**Case 2:** (Deep Equilibrium Model)

Very similarly,

$$\cos(\boldsymbol{h}_k, \boldsymbol{w}_k)_{\text{DEQ}} = \frac{\boldsymbol{h}_k^T \boldsymbol{w}_k}{\|\boldsymbol{w}_k\|\|\boldsymbol{h}_k\|}$$
$$= \frac{(\boldsymbol{I} - \boldsymbol{W}_{\text{DEQ}})\left(\boldsymbol{h}_k^0\right)^T \boldsymbol{w}_k}{\|\boldsymbol{w}_k\|\|\left(\boldsymbol{I} - \boldsymbol{W}_{\text{DEQ}}\right)^{-1}\boldsymbol{h}_k^0\|}$$
$$\geq \frac{2\left(\boldsymbol{I} - \boldsymbol{W}_{\text{DEQ}}\right)\left(\boldsymbol{h}_k^0\right)^T \boldsymbol{w}_k}{\|\boldsymbol{w}_k\|^2 + \frac{1}{2}\|\left(\boldsymbol{I} - \boldsymbol{W}_{\text{DEQ}}\right)\|^2 + \frac{1}{2}\|\boldsymbol{h}_k^0\|} \tag{61}$$
$$\geq \frac{4E_H\left(\boldsymbol{h}_k^0\right)^T \boldsymbol{w}_k}{1 + 2(E_W + E_H)(1 - E_H)}.$$

If the performance of DEQ is better than explicit neural network, then we have

$$\cos(\boldsymbol{h}_k, \boldsymbol{w}_k)_{\text{DEQ}} / \cos(\boldsymbol{h}_k, \boldsymbol{w}_k)_{\text{exp}} > 1,$$

which is equivalent to

$$\frac{E_H}{E_w + E_H} + E_H(1 - E_H) < 2. \tag{62}$$

In summary, though DEQ does not completely mitigate the issue of minority collapse, it shows significant improvement compared to explicit neural network under some conditions that are relatively easy to satisfy in the manifestation of the $\mathcal{NC}$ phenomenon.

# D   More experiments

In this section, we provide more experimental results, including the $\mathcal{NC}$ phenomena of Explicit NN and DEQ, and the training results under other imbalanced conditions.

Table 3: Test Accuracy on Cifar-10 and Cifar-100 Dataset with $K_A = 5$

| | | Cifar-10 | | | Cifar-100 | | |
|---|---|---|---|---|---|---|---|
| | $R$ | 10 | 50 | 100 | 10 | 50 | 100 |
| | overall | 80.73±0.48 | 63.08±0.87 | 44.86±1.43 | 52.62±0.86 | 41.62±0.68 | 37.33±2.29 |
| Explicit NN | majority | 94.18±0.56 | 91.02±0.89 | 89.32±0.79 | 74.10±1.03 | 73.94±0.25 | 74.24±1.13 |
| | minority | 67.80±0.35 | 35.14±0.65 | 0.40±3.86 | 31.10±0.70 | 9.30±1.10 | 0.42±3.04 |
| | overall | 81.36±1.03 | 65.03±1.90 | 46.09±1.77 | 53.31±0.98 | 44.07±2.04 | 39.11±2.46 |
| DEQ | majority | 93.14±1.81 | 90.88±2.83 | 90.20±0.85 | 72.90±1.65 | 75.98±1.75 | 75.79±0.96 |
| | minority | 69.58±0.66 | 39.18±1.46 | 1.26±4.93 | 33.72±0.79 | 12.16±3.75 | 2.42±5.89 |

Table 4: Test Accuracy on Cifar-10 and Cifar-100 Dataset with $K_A = 7$

| | | Cifar-10 | | | Cifar-100 | | |
|---|---|---|---|---|---|---|---|
| | $R$ | 10 | 50 | 100 | 10 | 50 | 100 |
| Explicit NN | overall | 83.17±0.40 | 66.91±0.39 | 53.27±0.81 | 59.11±0.84 | 51.71±1.02 | 50.72±0.60 |
| | majority | 89.09±0.36 | 80.90±0.57 | 75.12±0.74 | 71.93±0.65 | 72.20±0.66 | 72.46±0.58 |
| | minority | 69.37±0.49 | 34.26±0.30 | 2.30±1.01 | 29.20±0.92 | 3.90±1.29 | 0.00±0.00 |
| DEQ | overall | 83.78±1.85 | 69.47±1.86 | 56.74±0.98 | 60.51±0.88 | 52.99±1.86 | 51.79±0.92 |
| | majority | 88.98±1.99 | 82.91±2.22 | 78.81±0.67 | 72.90±1.19 | 72.99±0.98 | 73.98±0.66 |
| | minority | 71.65±1.63 | 38.12±1.61 | 5.20±1.91 | 31.13±0.83 | 6.33±2.35 | 0.00±0.00 |

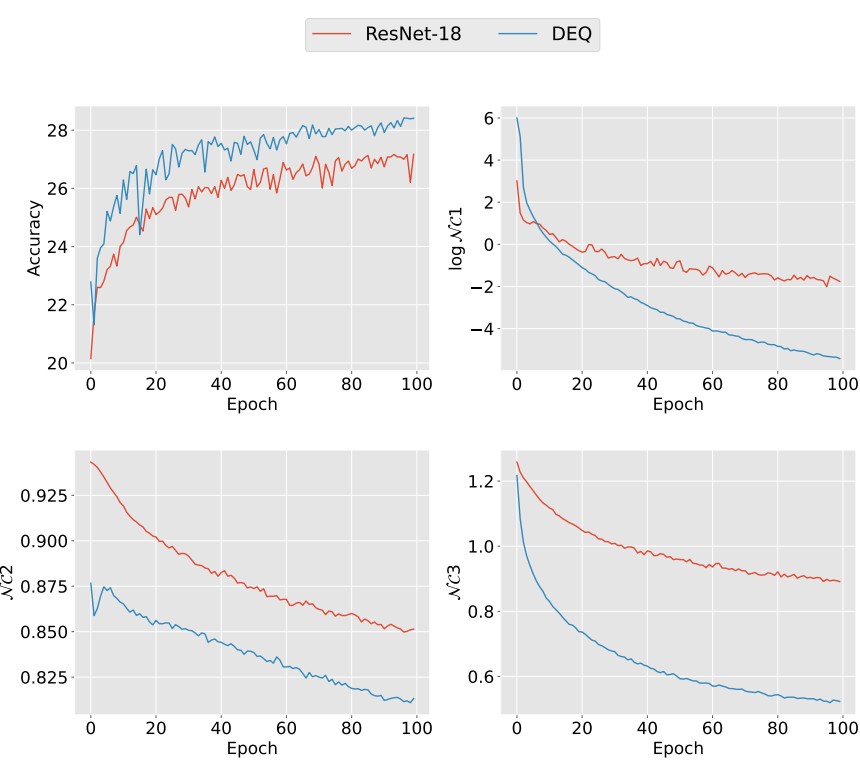

Figure 4: Accuracy and $\mathcal{NC}$ phenomenon on imbalanced dataset with $K_A = 3$, $K_B = 7$, $R = 100$

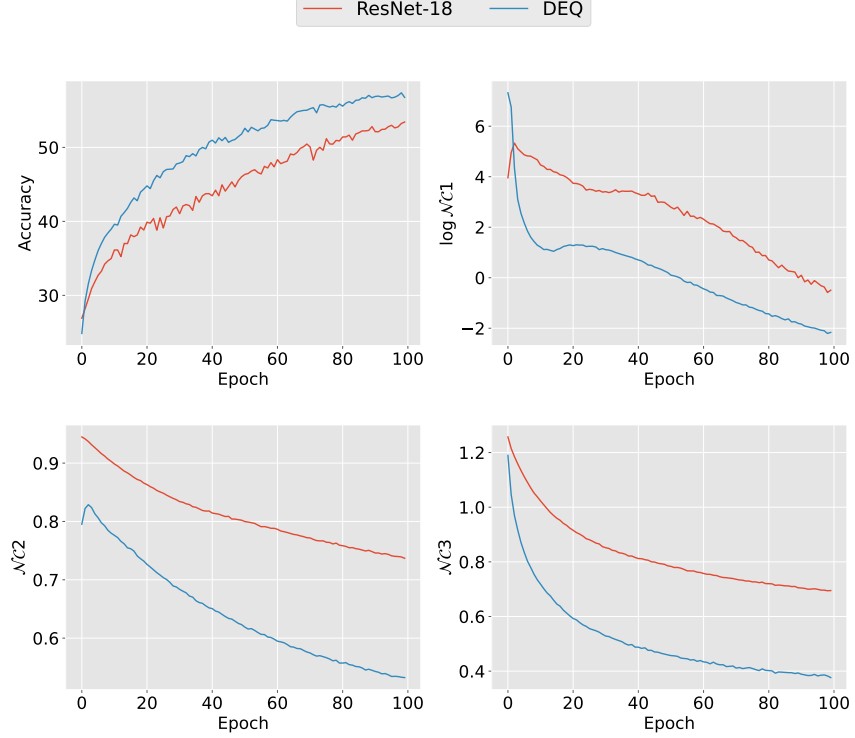

Figure 5: Accuracy and $\mathcal{NC}$ phenomenon on imbalanced dataset with $K_A = 7$, $K_B = 3$, $R = 100$

