# OpenReview forum: "Understanding Representation of Deep Equilibrium Models from Neural Collapse Perspective"
_NeurIPS.cc/2024/Conference — NeurIPS 2024 poster_

### Official Review · Reviewer_6ZyB · 2024-07-08

**Soundness:** 3
**Presentation:** 1
**Contribution:** 3
**Rating:** 5
**Confidence:** 4

**Summary:**

The paper analyzes the features of a deep equilibrium model (DEQ) using neural collapse metrics under class-balanced and imbalanced settings. In particular, it is shown that under imbalanced conditions, the class means of features in DEQ are relatively closer to a simplex ETF orientation than in the case of an explicit neural network model (such as ResNet).

**Strengths:**

The paper takes an interesting theoretical view of analyzing the features of deep equilibrium models using the neural collapse metrics. In particular, the paper extends the unconstrained feature model to cases with implicit layers and analyzes the settings under which neural collapse is ideal (with respect to data balancedness).

**Weaknesses:**

1. The paper suffers from major presentation issues. In particular, a lot of notation/formulation errors lead to unclear results. For instance:
- In Eq (1) and the text below, the function $\phi()$ is assigned to a matrix without mentioning the class-wise arrangement of the features in matrix $H$. Similarly, in the definition of NC1, the class means vector has a different notation when compared with the one presented in Appendix A (line 482).
- The NC2 formulation in Theorem 3.1 is wrong.
- $\tau(i)$ in Theorem 3.1 is not defined.
- Mismatch in the scale of NC1 values in Figure 3.

2. The author's claim about DEQ reaching lower loss when compared to explicit layers is not validated in the experiments. Especially, the (train) cross-entropy loss is not illustrated. Furthermore, based on the results in Figure 3, it is unclear when the ResNet/DEQ reaches the terminal phase of training (i.e 100% training accuracy).

3. The paper states that DEQ models can be memory efficient, but do not provide any numerical results on the computational benefits of employing them. Also, the experimental results rely on training with a single random seed which is an incomplete measure of good/bad performance.

**Questions:**

1.  The experimental setup with ResNet and the DEQ is unclear. Which part of the ResNet is being used as the image encoder? As of now, it seems like only the last ResNet block is reformulated as the DEQ, and all the previous layers are used as the image encoder. Is this observation correct? If that is the case, then what is the computational overhead of the DEQ layer?

2. Based on this setup of replacing only one ResNet block with the DEQ, how does this approach compare to simply fixing the last classifier layer of ResNet as a simplex ETF to address the class-imbalance issue?

3. What happens if we replace more than one block of the ResNet with a DEQ layer? How sensitive is the DEQ layer to learning rate and batch size?

---

> ### Author Rebuttal · Authors · 2024-08-07
>
> > Q1: The paper suffers from major presentation issues. In particular, a lot of notation/formulation errors lead to unclear results.
>
> A1: Thanks for pointing out these typos and errors! We have made the following corrections in the revised version:
> * In Line 74, we have adjusted the domain as $\phi(x):R^{in\times N}\rightarrow R^{D\times N}$.
> * We changed $\frac{E_WE_H}{N-1}$ on Line 549 to $-\frac{E_WE_H}{K-1}$, and we also adjusted the conclusion in NC2 from $k$ to $K$.
> * In Line 169, we replace $i \in \tau(i)$ with $i\in \tau (k)$，where $\tau(k)$ denotes the samples belonging to the $k$-th class.
> * We have adjusted the y-axis of Figure 3(a) for NC1 and changed it to a log scale for NC1.
>
>
> > Q2: The author's claim about DEQ reaching lower loss when compared to explicit layers is not validated in the experiments. Especially, the (train) cross-entropy loss is not illustrated.
>
> A2: Thanks for pointing this out! Since the specific value of the loss does not have practical significance in classification, we primarily use accuracy to compare the performance of explicit and implicit layers. In the revised version, we will include the presentation of loss values.
>
>
> > Q3: Based on the results in Figure 3, it is unclear when the ResNet/DEQ reaches the terminal phase of training.
>
>
> A3: As discussed in Section 5.1, this early-stop setting was intentional. This is because DEQ models exhibit instability, which becomes particularly pronounced as training progresses, with some samples struggling to converge to a fixed point and requiring more iterations. The previous paper raised this issue (section 3.1 in [1]) and addressed it using early stopping. Here, we follow this standard technique in DEQ.
>
> > Q4: The paper states that DEQ models can be memory efficient, but do not provide any numerical results on the computational benefits of employing them.
>
> A4: Memory efficiency is the fundemental properities of DEQ. Extensive experiments  have conducted on this aspect when it is proposed. According to table 1-3 in [2], for example, Transformer require 4.8 GB and 6.8 GB memory, while its corresponding DEQ version only requires 1.1 GB. This is because the core of DEQ lies in parameter weight sharing, which saves a large number of parameters. It is not the focus of our paper, so we did not include experiments on this. We primarily explain the performance of DEQ and compare it with explicit neural networks in classification problems.
>
>
> > Q5: The experimental results rely on training with a single random seed which is an incomplete measure of good/bad performance.
>
> A5: We would like to argue that this is not true. We did not use just one seed; instead, we used different random seeds and computed the average and standard deviation of accuracy, which are shown in the Table 1-4.
>
> > Q6: The experimental setup with ResNet and the DEQ is unclear. Which part of the ResNet is being used as the image encoder? As of now, it seems like only the last ResNet block is reformulated as the DEQ, and all the previous layers are used as the image encoder. Is this observation correct? If that is the case, then what is the computational overhead of the DEQ layer?
>
> A6: We are afraid that this is not true. In our experiment, we use the layers from the last stage of ResNet, specifically, the last two residual blocks, which comprise four neural network layers collectively structured as DEQ. This will reduce the number of parameter during computation. We apologize for any confusion in the previous statement and have clarified this in our revised paper.
>
> > Q7: Based on this setup of replacing only one ResNet block with the DEQ, how does this approach compare to simply fixing the last classifier layer of ResNet as a simplex ETF to address the class-imbalance issue?
>
> A7: Comparing DEQ to simplex ETF on the performance of imbalance learning is not appropriate. DEQ is a broader network representation and is not specifically designed to address imbalanced learning. It can incorporate any network structure as long as there are parameters to be learned. Thus, the last layer with a simplex ETF can also be converted into the DEQ formulation.
>
>
> > Q8: What happens if we replace more than one block of the ResNet with a DEQ layer?
>
> A8: Currently in our paper, we have already combined multiple consecutive residual blocks into DEQ. A distinctive feature of DEQ is its capability to convert arbitrary neural networks into the form of an implicit network regardless of length.
>
> > Q9: How sensitive is the DEQ layer to learning rate and batch size?
>
> A9: In our experiments, DEQ is not sensitive to batch size, but it is relatively more affected by the learning rate. A larger learning rate causes fluctuations in DEQ training to occur earlier. Due to the limited theoretical research on DEQ, these issues have not been thoroughly addressed and studied. Currently, [3,4] have explored different learning rate settings and compared their outcomes. [5] has proven that with sufficiently small learning rates, the loss function can converge at a linear rate. However, though we believe these are not within the scope of our paper, we will add the analysis into the related work part of our revised version.
>
> ref:
>
> [1] Stabilizing Equilibrium Models by Jacobian Regularization, icml 2021
>
> [2] Deep Equilibrium Models, nips 2019
>
> [3] Deep equilibrium networks are sensitive to initialization statistics, icml 2022
>
> [4] Positive Concave Deep Equilibrium Models, icml 2024
>
> [5] Global Convergence Rate of Deep Equilibrium Models with General Activations, arxiv: 2302.05797

---

> > ### Comment · Reviewer_6ZyB · 2024-08-11
> >
> > Thank you for clarifying some of the concerns.
> >
> > 1. Line 259 states that "In this context, accuracy is assessed by averaging the results from the last 10 epochs and computing their standard deviation". How does this relate to the random seed-based std dev results?
> >
> > 2. The author's response to Q3 regarding the terminal phase of training(TPT) in Figure 3 is a bit concerning. Since achieving TPT is the main underlying assumption for NC analysis, my major concern is that since the ResNet itself is not trained to achieve 100% training accuracy (TPT), the claims regarding better DEQ performance/ similar NC might not hold. The early stopping is only necessary to train the DEQ to achieve the best results, but isn't training data interpolation the primary requirement for NC?
> >
> > 3. The authors also mention that training a DEQ leads to training instabilities so they skip some samples (line 255). Can you provide some more context on how many samples are discarded in this fashion? Is it something to worry about?

---

> > > ### Author Response · Authors · 2024-08-11
> > >
> > > Thanks very much for your comments.
> > >
> > > A1. We are sorry for the misleading. Actually, we ran each experiment five times and then calculated the standard deviation. At each time, we average the results from the last 10 epochs and take this mean value as the result of this time. We didn't choose a specific random seed. During each training session, the results were based on PyTorch's default random seed settings. We will add this clarification in our final version.
> > >
> > > A2. We select 100 epochs as the early stopping criterion and achieve around 95% training accuracy generally for these experiments.  Since our work is primarily theoretical, our experiments aim at validating our theory and observing the trend of NC. DEQ with 95% TPT still has clear advantages in terms of NC values for imbalanced cases compared to the explicit counterpart with 100% TPT.  Therefore, we believe this result can validate DEQ’s advantage as claimed in our theory.
> > >
> > > A3. In our experimental results, using Cifar-100 as an example, less than 1% of the samples are discarded. Given the limited amount of affected data, we believe it is not quite necessary to worry about that.

---

> > > > ### Comment · Reviewer_6ZyB · 2024-08-11
> > > >
> > > > Thank you for the response.
> > > >
> > > > - I understand that training DEQ comes with instabilities and it is difficult to achieve TPT. However, I want to clearly emphasize that the NC metrics can be reduced even before reaching TPT (see Figure 6 in [1] ). However, the validity of neural collapse only comes into play once the network has interpolated the training data. So claiming that DEQ is better than its explicit counterpart (in terms of test performance) when the TPT conditions haven't been reached is not well justified in this context.
> > > >
> > > > - Also, the authors said in the A2 response above that *DEQ with 95% TPT still has clear advantages in terms of NC values for imbalanced cases compared to the explicit counterpart with 100% TPT*. What does this mean? Can the authors point me to experiments where they trained the explicit counterpart to 100% training accuracy? (Also please note that it is indeed possible to reach TPT with explicit networks under imbalanced conditions. (See Table 3 in [2]).
> > > >
> > > >
> > > > [1] Papyan, Vardan, X. Y. Han, and David L. Donoho. "Prevalence of neural collapse during the terminal phase of deep learning training." Proceedings of the National Academy of Sciences 117.40 (2020): 24652-24663.
> > > >
> > > > [2]  Fang, Cong, et al. "Exploring deep neural networks via layer-peeled model: Minority collapse in imbalanced training." Proceedings of the National Academy of Sciences 118.43 (2021): e2103091118. Arxiv link: https://arxiv.org/pdf/2101.12699

---

> > > > > ### Author Response · Authors · 2024-08-12
> > > > >
> > > > > Thanks very much for your further comments. We will first reply to Q2 and then Q1.
> > > > >
> > > > > > Q2: The authors said in the A2 response above that DEQ with 95% TPT still has clear advantages in terms of NC values for imbalanced cases compared to the explicit counterpart with 100% TPT. What does this mean? Can the authors point me to experiments where they trained the explicit counterpart to 100% training accuracy? (Also please note that it is indeed possible to reach TPT with explicit networks under imbalanced conditions. (See Table 3 in [2]).
> > > > >
> > > > > A2: Thanks for your comments. Of course, we agree that explicit neural networks say ResNet, can reach TPT under imbalanced cases. In our previous manuscript, we apply the same training setting (100 epochs) on both the implicit DEQ and explicit ResNet for a fair comparison.
> > > > >
> > > > > We conducted experiments that trained the explicit ResNet until 100 training accuracy, i.e. TPT. In the following table, we present the NC values of ResNet (100 epochs), ResNet (100% training accuracy, i.e. TPT), and DEQ (100 epochs) respectively. The parameters are set as $K_A = K_B = 5$ and $R = 100$.
> > > > >
> > > > > |     |   ResNet (100 epochs)   | ResNet (100% training accuracy, i.e. TPT) |  DEQ (100 epochs) |
> > > > > |:---:|:-------:|:-----:|:-----:|
> > > > > | NC1 |  0.0093  | 0.0045 | 0.0009 |
> > > > > | NC2 |  0.7980  | 0.7224 | 0.5312 |
> > > > > | NC3 |  0.7551  | 0.7365 | 0.5743 |
> > > > >
> > > > > Even though ResNet achieves 100% training accuracy, and DEQ (100 epochs) does not fully reach TPT (about 95% training accuracy), DEQ still shows a clear advantage compared to its ResNet in terms of NC values. Note that DEQ can be improved with some techniques as mentioned in [3,4] and may achieve TPT and enjoy even better NC performance; but this is out of our scope. We believe our experiment results are sufficient to show DEQ's advantage compared to its explicit counterpart. We will add these results and analyses in our final version.
> > > > >
> > > > > > Q1: I understand that training DEQ comes with instabilities and it is difficult to achieve TPT. However, I want to clearly emphasize that the NC metrics can be reduced even before reaching TPT (see Figure 6 in [1] ). However, the validity of neural collapse only comes into play once the network has interpolated the training data. So claiming that DEQ is better than its explicit counterpart (in terms of test performance) when the TPT conditions haven't been reached is not well justified in this context.
> > > > >
> > > > > A1: Thanks for pointing out this issue. We found that the results in [1] only show performance under balanced cases, leaving it uncertain whether the results still hold under imbalanced cases.
> > > > >
> > > > > After adding the above experiment results in A2, we believe DEQ's advantage can be verified. Besides, DEQ can be applied with some advanced techniques like monotone operators for more stable results, and some works [2-4] have already demonstrated that its accuracy can surpass traditional Explicit NN, with 100% training accuracy. But this is out of the scope of this paper. Since DEQ shares the same inner structure as standard neural networks, with the only difference being its solution methods, we have reasons to believe that if ResNet continues to improve, DEQ could also improve alongside it.
> > > > >
> > > > >
> > > > >
> > > > > Thank you again for pointing out these issues that help improve our paper quality. We hope that you can reconsider your score.
> > > > >
> > > > > Ref:
> > > > >
> > > > > [1] Prevalence of neural collapse during the terminal phase of deep learning training, PNAS.
> > > > >
> > > > > [2] Deep Equilibrium Models, NeurIPS 2019.
> > > > >
> > > > > [3] Monotone operator equilibrium networks, NeurIPS 2020.
> > > > >
> > > > > [4] Multiscale Deep Equilibrium Models, NeurIPS 2020.

---

> > > > > > ### Comment · Reviewer_6ZyB · 2024-08-12
> > > > > >
> > > > > > Thanks for presenting the experimental results.
> > > > > >
> > > > > > It is quite surprising that the NC1, NC2, and NC3 values of DEQ are lower than that of ResNet (with TPT), yet the DEQ can't reach TPT. What can be a possible explanation for that? In particular, how is it possible that in the case of DEQ, the feature class means are well-separated (lower NC1), and tend toward a simplex ETF (NC2), and yet the DEQ can't reach TPT? This seems counter-intuitive to the claim that they are better than simple ResNet's.

---

> > > > > > > ### Author Response · Authors · 2024-08-13
> > > > > > >
> > > > > > > We want to clarify the TPT issue in two scenarios. In balanced case, both Explicit NN and DEQ can reach TPT and achieve sufficient lower NC value. In the imbalanced case, both Explicit NN and DEQ may not show the neural collapse phenomenon even if the models reach TPT.  Therefore, we believe that our NC1-NC3 values do not necessarily indicate that the model has been trained to TPT. As shown in Figure 3, the NC values achieved by Explicit NN and DEQ are not sufficiently small to present the neural collapse. But we can still compare the NC values between DEQ and Explicit NN.
> > > > > > >
> > > > > > > Specifically, we would like to emphasize the following points in detail:
> > > > > > >
> > > > > > > * We do not claim "DEQ can't reach TPT". The reason DEQ does not reach 100% TPT is that under the current vanilla design of DEQ, it is challenging to avoid instability issues. As we analyzed before, integrating some advanced methods such as monotone operators can lead to sufficient training.
> > > > > > >
> > > > > > > * Our experimental results indicate that the test results at 100 epochs are not significantly different from those obtained 100% training accuracy. Since DEQ shares the same backbone as the corresponding Explicit NN, we believe that after reaching 100% training accuracy, DEQ can still exhibit better NC performance under imbalanced cases.
> > > > > > >
> > > > > > > * Further, the fixed-point iteration contributes to the better representation ability of DEQ. Our Theorem 4 provides theoretical evidence for DEQ's superior NC under imbalanced conditions.

---

> > > > > > > > ### Comment · Reviewer_6ZyB · 2024-08-13
> > > > > > > >
> > > > > > > > Thank you for the clarifications.
> > > > > > > >
> > > > > > > > I have increased my score, but I would urge the authors to please fix the presentation of the results and incorporate these discussions in the revisions. Especially the TPT aspect. Good luck.

---

> > > > > > > > > ### Author Response · Authors · 2024-08-14
> > > > > > > > >
> > > > > > > > > Thanks very much for your valuable comments and we will add these discussions in our final version.

---

### Official Review · Reviewer_Zj9t · 2024-07-10

**Soundness:** 3
**Presentation:** 3
**Contribution:** 2
**Rating:** 6
**Confidence:** 3

**Summary:**

This paper studies the neural collapse (NC) phenomenon in Deep Equilibrium model (DEQ), a competitive implicit neural network model to standard explicit model. The authors compare the theoretical property of DEQ and NN on the layer-peeled model, a simplified model that include the last two layers only. It was shown that under a balanced sample setting, NC happens both in DEQ and standard NN, under an imbalanced sample setting, DEQ exhibits better robustness property in terms of the NC metrics. Experimental results are presented to justify the theoretical analysis on CIFAR10 and CIFAR100, it was shown that DEQ have better performance than standard NN across in various settings.

**Strengths:**

The finding of this paper is novel and interesting, it provides a insightful perspective to understand the advantage of DEQ over the standard NN. The authors provide solid theoretical analysis and experimental results to support the main finding. Overall, the paper is well-organized and easy to follow.

**Weaknesses:**

My major concern of this paper is its contribution and significance. Since the discovery of minority collapse, there has been tons of literature on how to mitigate the minority collapse. For example, Proposition 1 in [1] proves that simply reweighting the samples can mitigate the minority collapse. Therefore, my opinion is that neural collapse analysis can provide limited insight about the advantage of DEQ over standard NN, since the improvement is relatively marginal compared with simple techniques. I would suggest the authors compare the DEQ and standard NN with some standard techniques such as reweighting, and confirm that the advantage indeed have practical influence.

[1] Exploring Deep Neural Networks via Layer-Peeled Model: Minority Collapse in Imbalanced Training

**Questions:**

1. The experimental results are limited to ResNet-18 only, I would encourage the authors to perform more extensive evaluation on various architectures to support the finding.

2. The proof needs to be better organized. It will be helpful if the authors can provide an outline of proof in the appendix, restate the main result and break it into several propositions, and add some high level intuition of why DEQ outperforms standard NN.

3. Neural Collapse solutions do not have full rank in general, would that bring issues when writing down the inverse of gradient in equation (3)?

4. In your theoretical analysis, it was assumed that the DEQ is linear with a specific expansion form. The authors should highlight this with a hold out assumption and discuss when it can be satisfied. In particular, what is $W_{\text{DEQ}^i}$ in line 162? Are they being optimized in the programming? In the current form it seems to reduce to simply a linear function $Wx$.

**Limitations:**

The authors have properly addressed the limitations.

---

> ### Author Rebuttal · Authors · 2024-08-07
>
> >Q1: My major concern of this paper is its contribution and significance. Since the discovery of minority collapse, there has been tons of literature on how to mitigate the minority collapse. Therefore, my opinion is that neural collapse analysis can provide limited insight about the advantage of DEQ over standard NN. I would suggest the authors compare the DEQ and standard NN with some standard techniques such as reweighting, and confirm that the advantage indeed have practical influence.
>
> A1: Thanks for your comment! We would like to clarify that our goal was not to use DEQ to mitigate minority class collapse but rather to leverage NC analysis to explain the performance of DEQ. We had employed reweighted CE to test both explicit NN and DEQ, observing that their accuracies both improved. Importantly, their difference remained similar. We shows the case of Cifar-10, $K_A=3$ and $R=10$, the results are as follows:
>
> ||Reweighted CE|CE|
> |:-:|:-:|:-:|
> |Explicit NN|79.01$\pm$0.36|72.57$\pm$0.25|
> |DEQ|80.17$\pm$0.95|73.84$\pm$0.72|
>
> Additionally, our work primarily focuses on theoretical aspects since current DEQ research heavily relies on empirical evidence for its effectiveness and performance and lacks theoretical grounding.
>
> Furthermore, we would like to argue that NC do not brings limited insight to DEQ. A significant similarity between DEQ and NC research is their model-agnostic nature. By using NC tools, we expand on the internal computational processes of both explicit and implicit structures, providing explanation of why DEQ might outperform explicit NN under certain conditions, i.e., we analyze and compare the differences between $W_{DEQ}$ and $W_{EX}$ in their application. Our findings demonstrate that the extracted features of DEQ align more closely with the vertices of a simplex ETF and exhibit better alignment with classifier weights under specific conditions.
>
> We quantify the benefits of DEQ’s multiple forward iterations, offering a valuable theoretical supplement to the existing research. Thus, we believe our paper contributes to the community by addressing a theoretical gap and providing a valuable addition to the existing research.
>
> >Q2: The experimental results are limited to ResNet-18 only, I would encourage the authors to perform more extensive evaluation on various architectures to support the finding.
>
> A2: Thanks for your advice! We conducted a new experiment using MobileNet as the network backbone. The results are as follows:
>
> $K_A=3$, Cifar-10
>
> ||R|10|50|100|
> |:-:|:-:|:-:|:-:|:-:|
> |Explicit NN|overall|69.47|49.16|34.98|
> ||major|94.13|95.56|93.70|
> ||minor|58.90|29.28|9.81|
> |DEQ|overall|71.12|49.88|35.59|
> ||major|95.24|96.00|94.01|
> ||minor|60.78|30.11|10.55|
>
> Due to character limits, we omitted the $\pm$ and variance term and selected a specific set of experiments to showcase. Additionally, we incorporated more results into our revised paper. The conclusion we found that is, the differences between Explicit NN and DEQ consistently persisted across different backbones.
>
> >Q3: The proof needs to be better organized. It will be helpful if the authors can provide an outline of proof in the appendix, restate the main result and break it into several propositions, and add some high level intuition of why DEQ outperforms standard NN.
>
> A3: Our proof sketch divides into two cases: in the balanced scenario, we utilize inequalities to iteratively establish conditions under which each equation holds, thereby deriving properties of NC. In the imbalanced case, we analyze conditions under which the majority and minority classes achieve their minima, and integrate these insights to compare the performance of Explicit NN and DEQ within the NC framework.
>
> We have re-orginzed our prove and made the following modification:
>
> - We add a table of contents for our appendix.
> - In the proof of balanced learning, we have rewritten Case 1 (explicit NN) in Line 528 and Case 2 (DEQ) in Line 552, emphasizing the three properties of NC for comparison. We explain that DEQ performs slightly better than explicit NN because of its lower bound on the loss function (theorem B.3).
> - In the proof of imbalanced learning, we have added more detailed explanations about why DEQ performs better under mild conditions, which is mainly because the forward fixed-point iteration increases the number of learning iterations for samples in the minority class.
>
> > Q4: Neural Collapse solutions do not have full rank in general, would that bring issues when writing down the inverse of gradient in equation (3)?
>
> A4: We don’t think so. Equation (3) describes the forward solving process of DEQ, where $x$ is the input to the DEQ layer, and  $z^\star$ the output of this layer through fixed-point iteration. The rank in (3) differs with that in NC. Besides, the matrix $B^{-1}$ is not directly obtained through inversion in practice, but rather approximated using BFGS.
>
> >Q5: In your theoretical analysis, it was assumed that the DEQ is linear with a specific expansion form. The authors should highlight this with a hold out assumption and discuss when it can be satisfied.
>
> A5: Thanks for your suggestion! We would like to clarify that the assumption of Linear DEQ, as used and discussed in [1], is a common approach in DEQ studies. This assumption posits that a linear function is used during the fixed-point iteration to compute the equilibrium point $z^\star$. We will add this assumption more clearly in the final version.
>
> >Q6: In particular, what is $W^i_{DEQ}x$ in line 162? Are they being optimized in the programming? In the current form it seems to reduce to simply a linear function $Wx$.
>
> A6: $W_{DEQ}$ represents the parameter being optimized, and $W_{DEQ}^i$ refers to its $i$-th power. Its presence arises from the necessity of repeated iterations during the forward solving process. It cannot be seen as a simple linear function $Wx$.
>
> ref:
> [1] Deep equilibrium networks are sensitive to initialization statistics, icml 2022

---

> > ### Comment · Reviewer_Zj9t · 2024-08-12
> >
> > Thank you for your detailed response and efforts on additional experiments, my concerns have been properly addressed. I would like to raise my score to weak accept and encourage the authors to improve the presentation of theoretical results in future versions.

---

> > > ### Author Response · Authors · 2024-08-13
> > > **Official Comment by Authors**
> > >
> > > Thanks so much for your insightful review and for raising your score. We are happy that our responses addressed your questions. We will take your suggestions into our revision.

---

### Official Review · Reviewer_a3Sj · 2024-07-11

**Soundness:** 3
**Presentation:** 2
**Contribution:** 3
**Rating:** 7
**Confidence:** 4

**Summary:**

This paper investigates the representation of Deep Equilibrium Models (DEQ), highlighting their memory efficiency and competitive performance. Using NC, it shows that DEQ exhibits NC under balanced conditions and maintains advantages in imbalanced settings. Theoretical findings are validated through experiments, demonstrating DEQ's superior handling of imbalanced datasets.

**Strengths:**

1 The paper provides a theoretical analysis of the representation of Deep Equilibrium Models under both balanced and imbalanced conditions using the Neural Collapse phenomenon. This analysis demonstrating DEQ’s advantages over explicit neural networks under some mild conditions.

2 Validation of theoretical insights through experimental results on datasets like Cifar-10 and Cifar-100, enhances the credibility of the findings that the superior performance of DEQ, especially in handling imbalanced datasets.

3 The introduction of the Neural Collapse phenomenon offers a novel perspective on deep representations in implicit neural networks.

**Weaknesses:**

1 The analysis is limited to simple imbalanced scenarios and DEQ models, restricting the generalizability of the findings to more complex real-world situations.

2 While the paper provides valuable theoretical insights and experimental validation in section 5, there may be a need for more in-depth quantitative analysis like statistical analysis to further support the claims and conclusions drawn.

3 The lack of broader experiment for $K_A\neq 3$ in Table 3 may hinder a comprehensive comparison of DEQ with Explicit NN.

**Questions:**

Please see the weaknesses.

**Limitations:**

The authors have adequately addressed the limitations.

---

> ### Author Rebuttal · Authors · 2024-08-07
>
> > Q1: The analysis is limited to simple imbalanced scenarios and DEQ models, restricting the generalizability of the findings to more complex real-world situations.
>
> A1: Thanks for your comment! Currently, our work primarily focuses on theoretical aspects, and discussing real-world issues will be a part of our future work. However, our research inherently addresses the model's generalization capability.
>
> Through our theoretical proof, we have established that DEQ's NC2 and NC3 demonstrate superior properties than explicit neural network under mild conditions. Specifically, we discovered that features extracted by DEQ exhibit closer alignment with the vertices of a simplex ETF and show better conformity with classifier weights under specific conditions. As a result, feature separability of DEQ is enhanced compared to explicit layers under some mild conditions, which can potentially lead to stronger generalization performance in downstream tasks.
>
>
>
> > Q2: While the paper provides valuable theoretical insights and experimental validation in section 5, there may be a need for more in-depth quantitative analysis like statistical analysis to further support the claims and conclusions drawn.
>
> A2: Thanks for your advice. We have plotted the values of NC1 - NC3 in Figure 3 to Figure 5 and their statistical representations are included in Appendix Section A. The NC cases shown in Figure 3(a) and Figure 3(b) precisely support the conclusions in section 3 and section 4 respectively. Additionally, we have also calculated the mean and standard deviation of training accuracy, which are presented in the Table 2 - Table 4.
>
> > Q3: The lack of broader experiment for $K_A\neq 3$ in Table 3 may hinder a comprehensive comparison of DEQ with Explicit NN.
>
> A3: We did conduct the experiments with $K_A$ values of 5 and 7, and the results were presented in the appendix (page 26).

---

> > ### Comment · Reviewer_a3Sj · 2024-08-09
> >
> > I appreciate the time and effort taken by the authors on the response to my review, and for addressing each of my concerns in turn. This is a well presented, thorough and novel piece of work. Moreover, this paper explores the connections and differences between implicit and explicit NNs.Two recent papers [p1, p2] bear relevance to the subject of this paper, and should be cited. Overall, I believe this work will be valuable for guiding future theory and practice in DEQs and other implicit networks. As such, I increase my rating to accept.
> >
> > [p1] X. Xie, et al. "Optimization induced equilibrium networks: An explicit optimization perspective for understanding equilibrium models." T-PAMI.
> >
> > [p2] Z. Ling, et al. "Deep Equilibrium Models are Almost Equivalent to Not-so-deep Explicit Models for High-dimensional Gaussian Mixtures." ICML 2024.

---

> > > ### Author Response · Authors · 2024-08-10
> > > **Official Comment by Authors**
> > >
> > > Thank you for your review and for appreciating our work! We will cite the two papers you mentioned in our final version.

---

### Official Review · Reviewer_C4pT · 2024-07-14

**Soundness:** 3
**Presentation:** 3
**Contribution:** 3
**Rating:** 5
**Confidence:** 2

**Summary:**

The author analyzes the DEQ from the prospective of Neural Collapse to demonstrate the reason why DEQ is effective. The Nerual collapse means that at the final phase of training (training error is close to zero), the feature and classifier vector converges to a simplex Equiangular Tight

Frame.  Neural collapse usually happens in balanced dataset since convergence in some classes in unbalanced dataset is difficult, which is called minority collapse.

To analyze neural collapse in DEQ, the author starts from a layer-peeled model, which focus on last-layer features. And they prove that Neural Collapse also happens in DEQ. However, the lower bound of loss function is smaller compared with explicit neural network.  In imbalanced dataset, the DEQ also show a smaller lower bound in loss function. One reason is that DEQ can be seen as an infinite layer network, which has better representation capacity. And the repeated iteration of DEQ can mitigate the problem in imbalanced dataset. However, minority collapse also happens in DEQ. The author provide some experiment to prove their claim.

**Strengths:**

1: To analyze the advantage of DEQ from perspective of DEQ is interesting. This paper may bring up some new perspective to analyze DEQ.

2: The author provide analysis on both balanced and imbalanced dataset.

3: The author provide extensive experiment to prove their idea.

**Weaknesses:**

1: The smaller lower bound of loss function may cause overfitting, which may affect the performance in the test phase.  Add discussions on this part can be helpful.

2: Maybe add some experiments on dataset other than Cifar can make the conclusion more persuasive.

**Questions:**

See weakness.

---

> ### Author Rebuttal · Authors · 2024-08-07
>
> > Q1: The smaller lower bound of loss function may cause overfitting, which may affect the performance in the test phase. Add discussions on this part can be helpful.
>
> A1: Thanks for your valuable suggestion! We agree with you that lower bound of loss function can lead to overfitting.
>
> We would like to claim that we did employ some manual setup methods to avoid overfitting. For example, we implemented an early stopping mechanism (sec 5.1) to prevent instability in DEQ and halt the training process before the model overfits the training data, thereby reducing the risk of overfitting.
>
> Additionally, the solving process of DEQ can be configured with specific thresholds to terminate at appropriate points, and a manual iteration limit can also be set.
>
> Furthermore, in our experiments, we also employed the Jacobian regularizer proposed in [1], which penalizes overly complex models and discourages the model from fitting the training data too closely.
>
> We have added the detailed discussion into our revised paper.
>
> > Q2: Maybe add some experiments on dataset other than Cifar can make the conclusion more persuasive.
>
> A2: Thanks for your comment. Following the settings in previous DEQ papers, we have also included results on additional datasets, such as SVHN and MNIST, the results are shown here:
>
>
> SVHN, $K_A=3$:
> |     |    R    |   10  |   50  |  100  |
> |:---:|:-------:|:-----:|:-----:|:-----:|
> | Explicit NN | overall | 83.82 | 60.66 | 54.84 |
> |     |  major  | 96.01 | 96.11 | 92.30 |
> |     |  minor  | 78.60 | 45.46 | 38.78 |
> | DEQ | overall | 85.14 | 64.00 | 56.52 |
> |     |  major  | 96.99 | 98.01 | 93.12 |
> |     |  minor  | 80.06 | 49.42 | 40.83 |
>
>
> SVHN, $K_A=5$:
> |     |    R    |   10  |   50  |  100  |
> |:---:|:-------:|:-----:|:-----:|:-----:|
> | Explicit NN | overall | 81.74 | 66.80 | 52.85 |
> |     |  major  | 91.33 | 88.86 | 84.41 |
> |     |  minor  | 72.16 | 44.73 | 21.30 |
> | DEQ | overall | 83.75 | 68.24 | 53.30 |
> |     |  major  | 93.52 | 90.90 | 86.12 |
> |     |  minor  | 73.98 | 45.58 | 20.48 |
>
>
>
> SVHN, $K_A=7$:
> |     |    R    |   10  |   50  |  100  |
> |:---:|:-------:|:-----:|:-----:|:-----:|
> | Explicit NN | overall | 82.36 | 71.41 | 63.97 |
> |     |  major  | 89.09 | 89.90 | 87.39 |
> |     |  minor  | 66.64 | 28.28 |  9.33 |
> | DEQ | overall | 82.63 | 72.17 | 65.21 |
> |     |  major  | 88.86 | 90.15 | 88.51 |
> |     |  minor  | 68.10 | 30.21 | 10.85 |
>
>
> And the trend also resembles as Cifar dataset. In the MNIST dataset, the results of DEQ and Explicit NN are quite similar due to the dataset's simplicity, so we do not present them here.
>
> ref：
>
> [1] Stabilizing Equilibrium Models by Jacobian Regularization, icml 2021

---

### Author Rebuttal · Authors · 2024-08-07

We thank the reviewers for their careful reading of our paper and help with improving our manuscript. We sincerely appreciate that you find our work:

- adds to the understanding of the behavior of DEQ (C4pT, a3Sj),
- conducts extensive and solid experiments (C4pT, a3Sj, Zj9t),
- addresses an novel, interesting and important topic (a3Sj, 6ZyB),
- provides rigorous proof and derives interesting conclusion (a3Sj, Zj9t, 6ZyB),
- is well-organized and easy to follow (Zj9t).

We would like to express our gratitude to all the reviewers for their valuable suggestions for improving our paper.

Several reviewers pointed out that if we add more experimental results could make the conclusion be more convincing. Therefore, we conducted additional experiments on **different datasets** other than Cifar-10 and Cifar-100, and also tested **alternative backbones** apart from ResNet. Interestingly, we obtained similar conclusions: expressing the network architecture in the form of DEQ provides a slight improvement in addressing imbalanced datasets.

Since there is currently a lack of related work analyzing the performance of DEQ, we would like to emphasize that our paper primarily focuses on analyzing the **theoretical explanation** for why DEQ outperforms explicit NNs in classification tasks under mild conditions. Furthermore, we are also **the first to consider the performance of DEQ on imbalanced datasets**.

Additionally, we also would like to highlight that incorporating NC into DEQ is motivated by NC's capability to analyze any form of neural network. The key feature of DEQ is also its ability to represent any network structure as an implicit model and solve it using fixed-point iteration. Both of them are **similar** in this aspect. Therefore we discussed the performance of parameters $W_{DEQ}$ and $W_{EX}$ for two different forms of networks using the NC tool. We observed that under certain mild conditions, DEQ tends to produce features that are closer to the simplex ETF. Moreover, there is better alignment between weights and features in DEQ.

In response to the issues raised by reviewers, we have implemented the following adjustments:

- Included the newly added experiments.
- Improved the textual explanations, especially added the analysis of the similarities between DEQ and NC.
- Conducted a thorough review of the proof process and add more detailed discussions.
- Corrected typos and addressed language expression errors.

---

### Author Response · Authors · 2024-08-10
**Please let us know if any further questions!**

Dear Reviewers and AC,

Thank you for providing valuable comments. We have tried to address the concerns raised by the reviewers.

We observe that, except for a3Sj, reviewers have no follow-up comments on our responses. Given the upcoming discussion stage deadline, please let us know if you have further questions. Besides, we would like to sincerely thank reviewer a3Sj for recognizing our contribution and raising their score from 5 to 7.

Thanks and best regards.

Authors

---

### Comment · Area_Chair_tSPn · 2024-08-13

Reviewer C4pT,

Could you please check the authors' responses and reply to them? The discussion deadline is coming soon!

Regards,

AC

---

### Decision · Program_Chairs · 2024-09-25

**Decision:**

Accept (poster)

**Comment:**

The authors present a systematic theoretical study of DEQ from the perspective of Neural Collapse. During the rebuttal phase, the authors actively discussed new experiments and clarifications with the reviewers. They properly addressed the concerns. Therefore, it should receive a clear acceptance.

Please follow the reviewers' suggestions to integrate the rebuttal content into the original paper.